

# URANOS v1.0 - the Ultra Rapid Adaptable Neutron-Only Simulation for Environmental Research

Markus Köhli[1,2], Martin Schrön[3], Steffen Zacharias[3], and Ulrich Schmidt[1]

[1]Physikalisches Institut, Heidelberg University, Heidelberg, Germany
[2]Physikalisches Institut, University of Bonn, Bonn, Germany
[3]Dep. Monitoring and Exploration Technologies, Helmholtz Centre for Environmental Research - UFZ, Leipzig, Germany

**Correspondence:** Markus Köhli (koehli@physi.uni-heidelberg.de)

**Abstract.**

The understanding of neutron transport by Monte-Carlo simulations led to major advancements towards precise interpretation of measurements. URANOS (Ultra Rapid Neutron-Only Simulation) is a free software package, which has been developed in the last years in a cooperation of Particle Physics and Environmental Sciences, specifically for the purposes of cosmic-ray neutron sensing (CRNS). Its versatile user interface and input/output scheme tailored for CRNS applications offers hydrologists a straightforward access to model individual scenarios and to directly perform advanced neutron transport calculations. The geometry can be modeled layerwise, whereas in each layer a voxel geometry is extruded using a two-dimensional map from pixel images representing predefined materials and allowing to construct objects on the basis of pixel graphics without a 3D editor. It furthermore features predefined cosmic-ray neutron spectra and detector configurations and allows also a replication of important site characteristics of study areas - from a small pond to the catchment scale. The simulation thereby gives precise answers to questions like: From which location do neutrons originate? How do they propagate to the sensor? What is the neutron response to certain environmental changes? In recent years, URANOS has been successfully employed by a number of studies, for example, to calculate the cosmic-ray neutron footprint, signals in complex geometries like mobile applications on roads, urban environments and snow patterns.

## 1 Introduction

The physical processes of neutron transport depend on the atomic composition of materials, on the individual neutron energy, and act across many orders of spatial scales. It is therefore not feasible to find generalized, analytical solutions under realistic conditions. Statistical and computational approaches are the only way to take all relevant physical interactions into account. In the so-called Monte Carlo codes millions of particles can be summoned with randomly sampled initial conditions, while their paths can be tracked and their interactions with nuclei obey the laws of physics. Finally, the summary statistics of those neutrons can reveal insights into their collective behavior. In the last decades, the Monte Carlo code MCNP6 (Goorley et al., 2012) and its predecessor MCNPX (Waters et al., 2007) were often consulted to study the behavior of neutrons near the surface (Desilets et al., 2006; Zreda et al., 2008; Franz et al., 2013; Zreda et al., 2012; Zweck et al., 2013; Desilets and Zreda, 2013; Andreasen et al., 2016). The conventional model accounts for all kinds of particles and various interactions, decreasing the computational



efficiency and resulting in complex model structures (and interfaces) which hamper a flexible use and are particularly difficult for new users to access. As an alternative for the growing user community of CRNS, we developed the Monte Carlo code URANOS (Ultra Rapid Adaptable Neutron-Only Simulation) which was specifically tailored to address open and recurring questions of the CRNS community (Köhli et al., 2015; Schrön et al., 2017; Schrön et al., 2018; Köhli et al., 2018b; Li et al., 2019; Schattan et al., 2019; Weimar et al., 2020; Köhli et al., 2021; Badiee et al., 2021; Francke et al., 2021). As the model has evolved, it has been proven to be useful for neutron spin echo detectors as in other research fields as well (Köhli et al., 2016, 2018a; Köhli and Schmoldt, 2021). URANOS is computationally very efficient as it only accounts for the most relevant neutron interaction processes, namely elastic collisions, inelastic collisions, absorption, and evaporation.

The main model features are: (1) tracking of particle histories from creation to detection, (2) detector representation as layers or geometric shapes, (3) voxel-based model extrusion and material setup based on color codes in ASCII matrices or bitmap images.

URANOS is designed as a Monte-Carlo tool which exclusively simulates contributions in an environment of neutron interactions. The standard calculation routine features a ray-casting algorithm for single neutron propagation and a voxel engine. The physics model follows the routines declared by the ENDF database standard and descriptions of implementations by OpenMC (Romano and Forget, 2013). It features the treatment of elastic collisions in the thermal and epithermal regime, as well as inelastic collisions, absorption and emission processes such as evaporation. Cross sections, energy distributions and angular distributions were taken from the databases ENDF/B-VII.1 (Chadwick et al., 2011), ENDF/B-VIII.0 (Brown et al., 2018) and JENDL/HE-2007 (Shibata et al., 2011). The entire software is developed in C++, linked against CERN's analysis toolbox ROOT (Brun and Rademakers, 1997), whereas the GUI uses the QT cross-platform framework.

## 1.1 Neutron Monte Carlo Codes

The Monte Carlo (N. Metropolis, 1949) method is a brute-force calculation technique, which is used for complex problems consisting of well-defined and/or independent sub-tasks. The method solves a problem by repeated random sampling from a set of initial conditions and interactions. Among the existing Neutron Monte Carlo tools, most codes have a long history and strong aim towards nuclear fuel calculations. Besides dedicated programs, the most widely used in neutron physics is MCNP, followed by GEANT4 (S. Agostinelli, et al., 2003).

MCNP (**M**onte **C**arlo **N-P**article) was developed as a general purpose software to treat neutrons, photons, electrons and the coupled transport thereof, excluding magnetic field effects. Versions until MCNP4 (Briesmeister et al., 2000) were written in FORTRAN 77 (Sun Programmers Group, 1995a), which until the mid-90s was considered the standard in scientific computing. MCNP4 is capable of simulating neutrons up to 20 MeV, which is the maximum of most of the cross sections available in the evaluated data bases. With version 5 (X-5 Monte Carlo Team, 2003) the development was forked to the MCNPX (Waters et al., 2007)(MCNP e**X**tendend) branch, which converted the code to Fortran 90 (Sun Programmers Group, 1995b) and included the LAHET (Prael and Lichtenstein, 1989) framework. This especially introduced the extension of the energy range for many isotopes up to 150 MeV and some to several GeV by using the Cascade-Exciton Model (CEM) (Gudima et al., 1983) and ontop the Los Alamos Quark-Gluon String Model (LAQGSM) (Gudima et al., 2001). It can also treat (heavy) ion transport for



charged particles with energies larger than 1 MeV/nucleon by tabulated interaction ranges. The actual version 6 (Goorley et al.,
2012) re-merged the X-branch into the main development branch. It provides an optional cosmic-ray source (McKinney et al.,
2012) which can be used to produce a cosmic neutron spectrum (McKinney, 2013).

A more recent general purpose tool is PHITS (Iwase et al., 2002) (**P**article and **H**eavy **I**on **T**ransport code **S**ystem), as extension of the high energy particle transport code NMTC/JAM (Niita et al., 2001), which, besides the features mentioned above, also supports charged particles in magnetic fields, $\mathrm{d}E/\mathrm{d}x$ calculations in the **C**ontinuous-**S**lowing-**D**own **A**pproximation (Nelms, 1956) (CSDA) and intra-nuclear cascade (JAM) (Niita, 2002) (**J**et **AA** **M**icroscopic Transport) models up to 1 TeV. PHITS is typically linked against the JENDL-4/HE(High Energy) and later data bases, consisting of files evaluated by CCONE (Iwamoto et al., 2016), which is a more sophisticated model compared to INCL (Boudard et al., 2013) and JAM. It comes along with many adjustable parameters for each nucleus, which often leads to a better accuracy compared to other physics models. One of the recent follow-up developments is PARMA (Sato et al., 2008) (**P**HITS-based **A**nalytical **R**adiation **M**odel in the **A**tmosphere). It calculates the spectra of leptons and hadrons providing effective models for fluxes of particles of different species, especially with the aim of dose estimations.

The FLUKA (Battistoni et al., 2015)(**FLU**ktuierende **KA**skade) code is mostly oriented towards charged hadronic transport and nuclear and particle physics experiments. For neutron calculations, the full spectrum is divided into 260 energy groups, which are not directly linked to an evaluated data base, but operate on their own set of reprocessed and simplified mean values. Especially for neutrons and geometrical representations, it contains reimplementations from the MORSE (Emmett, 1975) neutron and gamma-ray transport code.

GEANT4 (S. Agostinelli, et al., 2003) (**GE**ometry **AN**d **T**racking) can be regarded as FLUKA's successor, based on multithreaded C++ and OpenGL visualizations. It is designed specifically for the needs of high-energy and accelerator physics. GEANT4 especially excels in describing complex geometries. Since 2011, also driven by requests from the European Spallation Source (Peggs et al., 2013), an increasing number of low-energy neutron calculation features were introduced. Meanwhile the software has advanced to a level where there is a good agreement with other codes like MCNP for fast neutrons (Solovyev et al., 2015) as well as slow neutrons (van der Ende et al., 2016).

## 1.2 Why another code?

The choice for creating an own independently operating Monte Carlo based program apart from the mentioned codes was based on evaluating the specific demands of understanding the physics of neutron detectors. The key ideas are:

- Most of the existing codes are not publicly available and fall under the export control law for nuclear related technology - whereas the underlying data bases are free of access. High precision detector development is not a use case which is envisaged by the authorities.

- Most of the existing codes were developed in the 1970s or 1980s. Written in the procedural programming language Fortran, which has been proven useful in the ages of limited execution orders and memory, these tools nowadays suffer the drawback of requiring sophisticated and time-consuming code tuning. In the best case they received wrappers in



C, rarely in C++. Today, facing multithreading, distributed network services and distributed memory in abundance, the changes of computing technology also have a strong impact on the code design and coding strategies.

– Meanwhile even more complex mathematical operations are readily available from standard packages like the GSL (Galassi et al., 2016) (**G**NU **S**cientific **L**ibrary) and frameworks such as ROOT (Brun and Rademakers, 1997).

– The majority of codes focuses on the evaluation of radiation sources, including gamma emissions. For example, signal generation in a boron-based hybrid detector requires two additional steps of charged particle transport mechanisms - within the conversion layer itself and subsequently in the gas. In the most cases it is not possible to integrate such a calculation path directly, but it would have to be added on top of the simulation. Furthermore, typical codes expect for the geometry objects of roughly equal size - boron layers having an aspect ratio of $10^5$ due to the low thickness are computationally difficult to model.

– All available codes propagate a take-off amount of neutrons in time as in typical applications like criticality calculations the neutrons themselves change the state of the environment, for example by generating a significant amount of heat. Therefore, the whole ensemble has to be propagated in time, especially until an equilibrium state is reached. Due to limited computing resources this required simplifications like the multigroup method.

– The multigroup method is a technique, which allows significant improvement of the calculation speed by not treating every neutron track individually but assigning an effective weight to propagating particle. This weight gets increased for production processes and reduced, if a neutron is absorbed or loses enough energy to drop out of a specific interval. The method is derived from solving Fermi age diffusion equations (Hébert, 2010) and is applied in many codes. However, it requires many interactions to generate enough randomness and thus it leads to a significant bias in situations when neutron will likely undergo only 1–2 collisions. For the study of background contributions in detectors or albedo neutrons, such a systematic error should be avoided.

The only software package which does not suffer from the mentioned drawbacks was GEANT4. Yet when the work on URANOS started in 2014, the GEANT4 code did not at all feature any accurate low energy neutron calculation. Materials in GEANT4 are usually described under a free gas assumption with unbound cross sections with no information about interatomic chemical bindings. This especially comes into play when treating hydrogen collisions - GEANT4 though can be coupled to the constantly developed models for evaluating the JEFF-3.X (Koning et al., 2011) ACE formatted thermal scattering law files. For scattering in crystal structures meanwhile the NXSG4 extension (Kittelmann and Boin, 2015) has been released, which reduced the amount of relevant physics necessary to be integrated. Still, the performance of GEANT4 in typical scenarios is significantly lower than those of other codes. In conclusion it has been decided to focus on a design from scratch in a modular, object oriented language.





## 2 Calculation routines

### 2.1 Sampling

The Monte Carlo approach is a stochastic method, in which properties of generated neutrons are randomly chosen from a
predefined probability distribution. Examples for sampled neutron transport properties are: (a) the path length, sampled from
the probability $p$ of an interaction on a distance $\mathrm{d}x$ in a homogeneous material, $l = -\ln(r)/\Sigma$, or (b) the thermal neutron
velocity distribution.

### 2.2 Random number generation

The pseudo-random number generator `TRandom3` uses the Mersenne-Twister algorithm MT 19937 (Matsumoto and Nishimura,
1998) based on the Mersenne prime number 19937. It has a very long period of $p = 2^{19937} - 1 \approx 4.3 \cdot 10^{6001}$, low correlation
between subsequent numbers (k-distributed for the output sequence) and is relatively fast, as it generates the output sequence
of 624 32 bit integers at once - `TRandom3` takes approximately 10 ns for each random number on a modern architecture. The
generator is seeded at the initialization of the program by the system time in milliseconds. This timestamp is taken as the first
integer of the seed sequence, the remaining 623 numbers are generated by the multipliers from Knuth (1997).

### 2.3 Sampling free path length

The probability $p$ of an interaction along a distance $\mathrm{d}x$ in a homogeneous material can be described as

$$\mathrm{d}p = \Sigma_t \mathrm{d}x \tag{1}$$

with the macroscopic cross section $\Sigma_t$, which in general is energy dependent. Solutions of this type of differential equation are
exponential functions. For the non-interaction probability one therefore can write

$$p(x) = \exp(-x\Sigma_t). \tag{2}$$

The probability distribution function for the distance to the next collision (2) assuming conditional probabilities transforms to

$$p(x)\mathrm{d}x = \Sigma_t \exp(-x\Sigma_t)\,\mathrm{d}x. \tag{3}$$

The free path length $l$ is obtained by the cumulative probability distribution function of (3) by

$$\int_0^l p(x)\mathrm{d}x = \int_0^l \Sigma_t \exp(-x\Sigma_t)\,\mathrm{d}x = 1 - \exp(-\Sigma_t l) = P(l). \tag{4}$$

In order to retrieve a path length, (4) can be sampled using the inversion method. This means, that the normalized cumulative
function is set equal to a random number $\xi$ on a unit interval:

$$l = -\frac{\ln(1-\xi)}{\Sigma_t} = -\frac{\ln(\xi)}{\Sigma_t}. \tag{5}$$





As $\xi$ is uniformly distributed in $[0,1)$ the same holds true for $1-\xi$, justifying the latter transformation.

It is assumed in (5) that the material is homogeneous and the cross section along with the kinetic energy stay constant. In
case of an inhomogeneous material it is possible that the integral cannot be resolved in a closed form. The solution is to split
the domain into entities of homogeneous materials and only evaluate the path to the respective border. This procedure is equal
to the prerequisite already stated in (3), that the probability at any point $x$ does not depend on the individual path history.

### 2.4 Sampling thermal velocity distributions

Scattering processes with thermal neutrons require, besides sampling a Maxwell-Boltzmann distribution, that it has to be taken
into account that relative velocities with respect to the target nuclide have an influence on the cross section, i.e. reaction rate.
An algorithm has to be applied which preserves the thermally-averaged reaction rate. Such has been introduced by Gelbard
(1979), whereas this modified version follows the implementation by Romano and Forget (2013) and Walsh et al. (2014). By
using the effect of thermal motion on the interaction probability

$$v\overline{\sigma}(v,T) = \int v_r \sigma\left(v_r\right) f_M^{(1)}(V) \mathrm{d}\boldsymbol{V}, \tag{6}$$

one has to conserve the reaction rate, the integrand of (6),

$$R(V) = \|\boldsymbol{v}-\boldsymbol{V}\|\sigma\left(\|\boldsymbol{v}-\boldsymbol{V}\|\right) f_M^{(1)}(V), \tag{7}$$

whereas $f_M^{(1)}(V)$ denotes the velocity distribution for target nuclei of temperature $T$, velocity $\boldsymbol{V}$ and magnitude of veloc-
ity $V$. The center-of-mass (CM) system of the collision of a neutron with velocity $\boldsymbol{v}$ moves at $v_r = \|\boldsymbol{v_r}\| = \|\boldsymbol{v}-\boldsymbol{V}\| = \sqrt{v^2+V^2-2vV\cos\vartheta}$. Such a probability function can be constructed by

$$p(V)\mathrm{d}V = \frac{R(V)\mathrm{d}V}{\int R(V)\mathrm{d}V}. \tag{8}$$

Defining the denominator of (8) as the normalization factor $C$ and

$$\beta = \sqrt{\frac{m}{2k_B T}} \tag{9}$$

as well as $\mu = \cos\vartheta$ one obtains

$$p(V,\mu)\mathrm{d}V\mathrm{d}\mu = \tag{10}$$

$$\frac{4\sigma(v_r)}{\sqrt{\pi}C'}\sqrt{v^2+V^2-2vV\mu}\beta^3 V^2 \exp\left(-\beta^2 V^2\right)\mathrm{d}V\mathrm{d}\mu.$$

In order to obtain a sampling scheme one can divide (10) into two parts such that

$$p(V,\mu) = g_1(V,\mu)g_2(V) \tag{11}$$

$$g_1(V,\mu) = \frac{4\sigma(v_r)}{\sqrt{\pi}C'}\frac{\sqrt{v^2+V^2-2vV\mu}}{v+V}$$

$$g_2(V) = (v+V)\beta^3 V^2 \exp\left(-\beta^2 V^2\right).$$





Here the reason for dividing and multiplying (10) by $v + V$ is that $g_1$ is bounded. As $\|v - V\|$ can take on arbitrarily large values, dividing by the sum of the speeds as the maximum value ensures it to be bounded. In general a probability distribution function $q(x) = g_1(x)g_2(x)$ can sampled by sampling $x'$ from a normalized distribution $q(x)$

$$q(x)\mathrm{d}x = \frac{g_2(x)}{\int g_2(x)} \tag{12}$$

and accepting it with a probability of

$$p_{\mathrm{accept}} = \frac{g_1(x')}{\max[g_1(x)]}, \tag{13}$$

with $g_1(x)$ bounded. In order to determine $q(V)$ it is necessary to integrate $g_2$ into (11)

$$\int\limits_0^\infty \mathrm{d}V (v + V)\beta^3 V^2 \exp\left(-\beta^2 V^2\right) = \frac{1}{4\beta}\left(\sqrt{\pi}\beta v + 2\right), \tag{14}$$

leading to sampling the probability distribution function

$$q(V)\mathrm{d}V = \left(\frac{4\beta^4 v V^2}{\sqrt{\pi}\beta v + 2} + \frac{4\beta^4 V^3}{\sqrt{\pi}\beta v + 2}\right)\exp\left(-\beta^2 V^2\right). \tag{15}$$

By substituting $x = \beta V$, likewise $\mathrm{d}x = \beta \mathrm{d}V$, and $y = \beta v$ leads finally to

$$q(x)\mathrm{d}x = \left[\left(\frac{\sqrt{\pi}y}{\sqrt{\pi}y + 2}\right)\frac{4}{\sqrt{\pi}}x^2 \exp\left(-x^2\right)\right. \tag{16}$$
$$\left. + \left(\frac{2}{\sqrt{\pi}y + 2}\right)2x^3 \exp\left(-x^2\right)\right]\mathrm{d}x.$$

The terms outside the parentheses are normalized probability distribution functions which allow to be sampled directly and the expressions inside the parentheses are always $< 1$.

The thermal neutron scattering sampling scheme therefore is the following:

A random number $\xi_1$ is sampled from $[0, 1)$ and if

$$\xi_1 < \frac{2}{\sqrt{\pi}y + 2}, \tag{17}$$

the function $2x^3 \exp\left(-x^2\right)$ is sampled, otherwise the function $4/\sqrt{\pi}x^2 \exp\left(-x^2\right)$. The retrieved $x$ gives the value for $V$ by dividing by $\beta$. The decision to accept this velocity is based on (13). The cosine of the angle can be sampled by another random

number $\xi_2$ in $[0, 1]$ by

$$\mu = 2\xi_2 - 1 \tag{18}$$

and as the maximum of $g_1$ is $4\sigma(v_r)/\sqrt{\pi}C'$ another sampling random number $\xi_3$ can be used to accept speed and angle by

$$\xi_3 < \frac{\sqrt{v^2 + V^2 - 2vV\mu}}{v + V}. \tag{19}$$

If this condition is not met, speed and cosine of the angle have to be resampled.





## 2.5 Nuclear data

Experimental and theoretical results on neutron-nuclear interactions and their subsequent products are collected in libraries. The main data base is the Experimental Nuclear Reaction Data Library (EXFOR) (Otuka et al., 2014), which stores most of the accepted published results. Results of neutron interaction measurements are sometimes contradictory and often not comprehensive, therefore so-called evaluated data bases have been created, which assess the literature especially regarding the intercomparison of different results and compress them to standardized values. The data bases, which are used for this work are the United States Evaluated Nuclear Data File (ENDF/B) and the Japanese Evaluated Nuclear Data Library (JENDL). Especially the high energy branch JENDL-HE provided the largest neutron interaction data set relevant for environmental studies. The ENDF format uses MT numbers to define reaction types and MF numbers to classify the data type of the respective set (Trkov et al., 2012).

## 3 Model Design

The design of URANOS was motivated by the following general aspects:

- The geometry is represented in a three-dimensional coordinate space with dimensions from the centimeter to the kilometer scale.

- In typical model runs the number of neutrons can easily reach $10^9$ with only one in a million neutron contributing to an observable. For this reason ensemble statistics would not be applicable.

- The relevant interactions are typically not deterministic but of statistical (random) nature.

- Important parameters like cross sections cannot be derived analytically but have to be extracted from tabulated data bases.

- Often neutron interactions can be reduced to a subset of relevant interactions, predominately of not more than two different types.

- Most particles other than neutrons are typically not contributing, while URANOS is still capable of modeling consecutive conversion ions necessary for signal generation.

- For the creation and propagation of high-energy neutrons within particles cascades effective models can be applied.

### 3.1 Geometrical layer concept

One specific feature of URANOS is its layer geometry, which takes advantage of the lateral symmetry of typical modeling problems, may it be an air-ground interface or the buildup of a neutron detector. The concept is presented in Fig. 1. Whereas along the horizontal and vertical axes the geometric scales vary significantly, the mean free path lengths in both directions





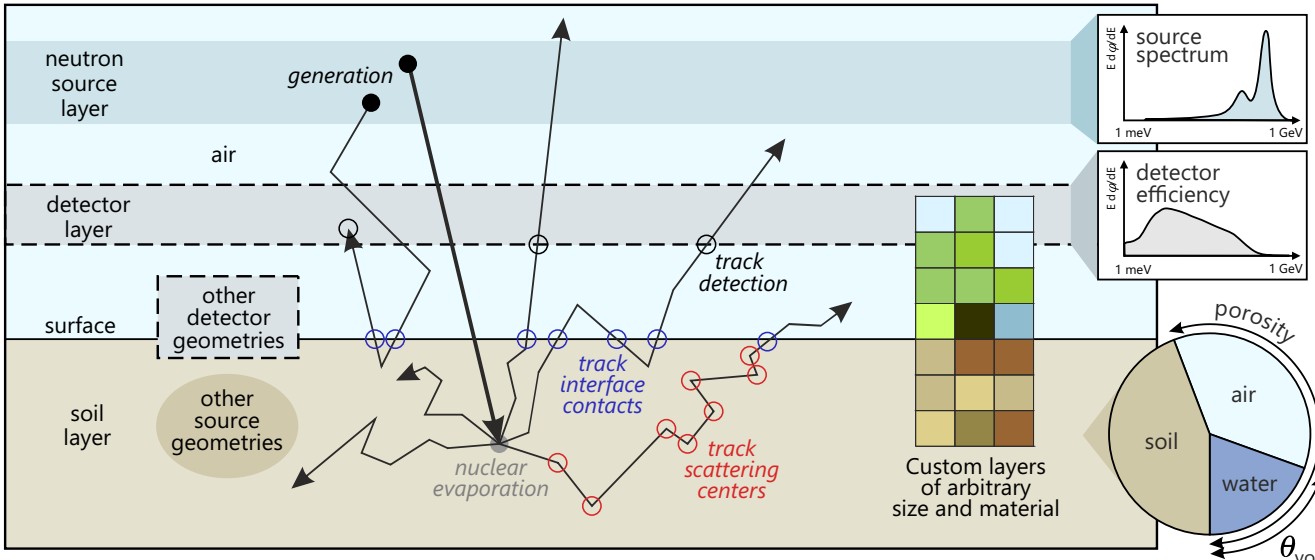

**Figure 1.** Two-dimensional projection of the 3D model setup and processes. Neutrons are generated in a source layer following a source spectrum that resembles the energy distribution of particles propagated through the air column above. High-energy neutrons create fast neutrons by nuclear evaporation in the soil. They further scatter through the material layers until they eventually reach the detector layer which absorbs the neutrons depending on the chosen detector response function. While the soil is modeled as a combination of silicate, air, and water, various other components of the environment can be modeled with the available material and density options.

are comparable with respect to the spatial extension. For example the absorption probability for a neutron in a 500 nm film of boron might be around 3 %, the scattering probability in a polymer foil of 100 times the thickness is approximately the same number and in the air gap of 100 times the thickness of the plastic it may be 0.3%. This also means that these large differences in the spatial dimensions are a challenge in terms of defining the geometries for simulation. The solution of URANOS is using layers. This allows to easily build a geometry of homogeneous materials with the main parameter being position and height of such a layer. Each layer furthermore can be sub-structured by two-dimensional matrices into voxels. Applying periodic or reflecting boundary conditions to the domain can improve the statistics or reduce the effort for building the simulation model.

## 3.2 Ray casting

In contrast to other Monte Carlo models focusing on fuel calculations, URANOS uses the method of ray-casting in order to keep track of the particles. This improves the accuracy in cases where only a specific subset of conditions will meet the criteria for scoring. The ray-casting technique (Roth, 1982) refers to conducting a series of ray-surface intersection tests in order to determine the first object crossed by tracks from a source. These intersections are either defined by analytical surfaces, like the layer structure, or computed from extruded voxels, which do not at all consist of surfaces. Similar types of geometry definitions with mixed volume and surface data were for example used in early computer games when no powerful hardware





acceleration was available and nowadays for X-ray tomography image reconstruction in material research (Maire and Withers, 2013), geosciences (Cnudde and Boone, 2013) and especially medical imaging (Goldman, 2007). The method of ray casting also allows to only record and store the variables necessary for each run. The neutron is propagated forward in time through the

domain and flags are used as boolean operators for each possible output. If for example the recording observable is defined as the density above the surface not the whole track but only the tracklet within the layer above the ground is kept in the memory.

## 4 Computational structure

The basic concept of URANOS relies on looping over a set of neutrons, which features initial conditions, predefined or randomized, and on loop tracking of the path of each neutron through the geometry. Both entities are referred to as 'stacks'. In

each step the geometrical boundaries are determined and handed over to the physics computation unit. For specific cases actual variables of the neutron or its track history are recorded emulating a real or a virtual detector. This process is called 'scoring' and can be invoked when passing a layer or an absorption in a converter takes place. A track is defined as the shortest path between two points of interaction. As will be seen later, it can be cut by layer or material boundaries, which dissects it into tracklets. The workflow shown in Fig. 2 illustrates the entire simulation process, which will be described in the following.

### 4.1 Startup

Before the main calculation routine three steps are carried out:

– Assigning memory to objects, which will be used throughout the calculation, by creating empty containers. These are at least 50 one- and two-dimensional root histograms.

– Reading the configuration files, creating the geometry and, if available, reading the voxel extrusion matrices.

– Reading the necessary tables from the ENDF library and loading them into the system memory.

The configuration is split into two files, one containing the basic settings for URANOS, like the number of neutrons to calculate and furthermore import and export folders for the data, and one containing information how to geometrically structure the layers, see here also the next sec. 4.2. Cross sections and angular distributions are read from tabulated ENDF files, exemplarily shown in Fig. 3, grouped into absorption, elastic and inelastic scattering. Tab. 1 shows exemplarily the selected cross sections to be loaded for $^1$H, $^{10}$B and $^{16}$O, whereas the full list of available isotopes can be found in appendix A. Only MT numbers

**Table 1.** Example cross sections according to the ENDF standard..

|  | Elastic | Inelastic | Absorption |
| --- | --- | --- | --- |
| $^1$H | MT=2 | n/A | MT=5, 102, 208–210 |
| $^{10}$B | MT=2 | MT=51–54 | MT=107 |
| $^{16}$O | MT=2 | MT=51–70 | MT=5, 102, 103, 107, 208–210 |



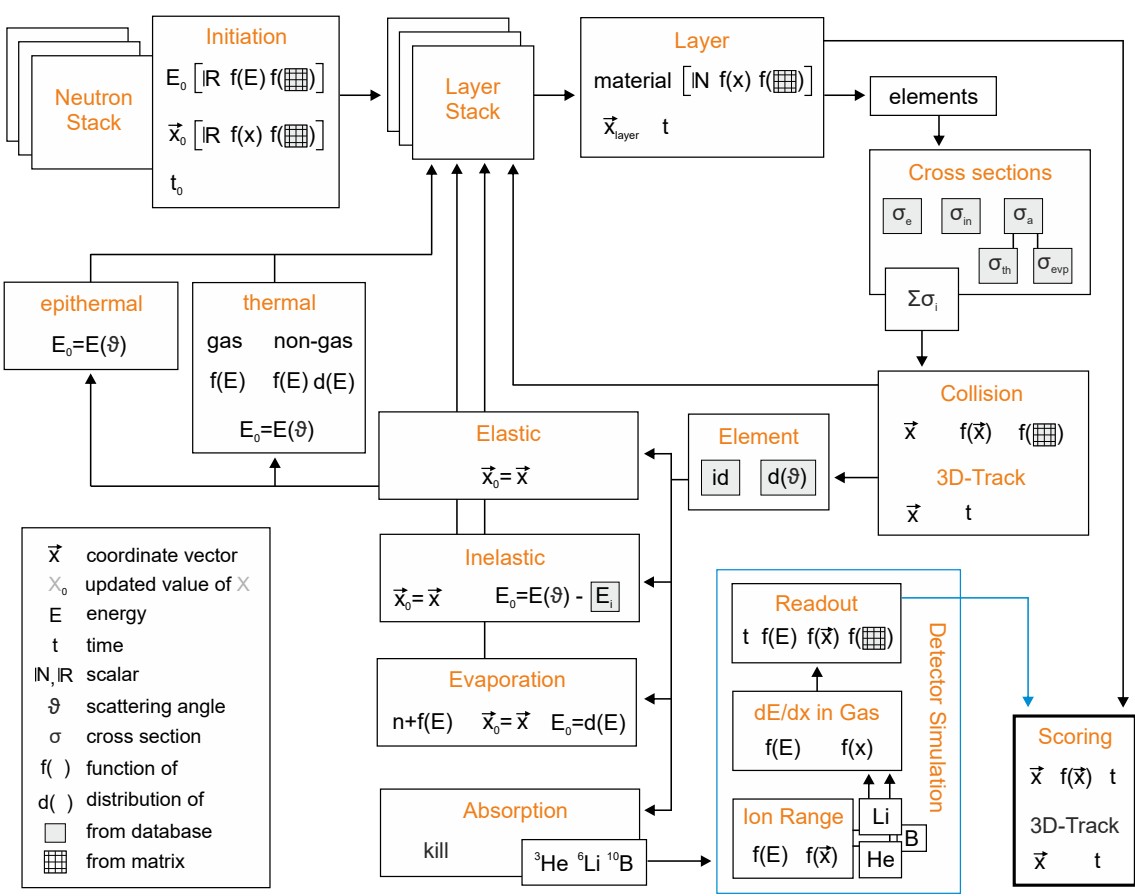

**Figure 2.** Internal workflow of URANOS. Each calculation step is represented by a block describing the structural function in orange and the corresponding physics variables.

with significant contributions are taken into account, which translates to omitting processes with overall less than $10^{-2}$ % of the total cross section. Furthermore, the cross section tables are compressed before loaded into the memory. Except for hydrogen, the algorithm skips every consecutive value with a relative difference of less than 1 % to its non-skipped predecessor, removing 0 % (rare elements) to 98 % (iron) of data, which saves a significant amount of iteration steps of determining the cross section.

The smallest error listed on cross sections can be found for elastic scattering of hydrogen with 0.3 %, other isotopes exhibit standard deviations of 1 % and larger, which justifies the compression method. For calculating the total macroscopic cross section the individual contributions of elastic $\Sigma^{\mathrm{e}}$ and inelastic $\Sigma^{\mathrm{in}}$ scattering as well as absorption $\Sigma^{\mathrm{a}}$ are summed up

$$\Sigma_t = \Sigma^{\mathrm{a}} + \Sigma^{\mathrm{e}} + \Sigma^{\mathrm{in}}, \tag{20}$$

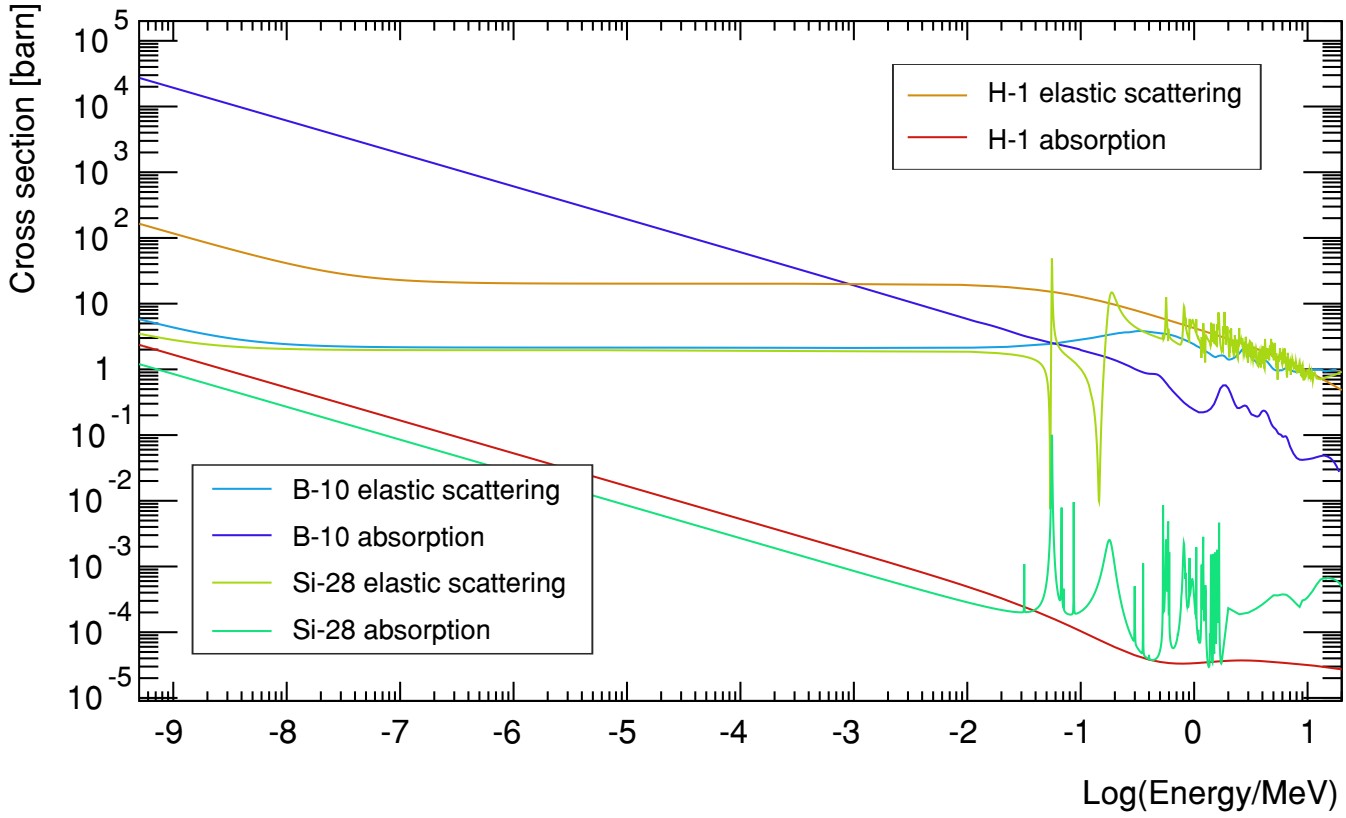

**Figure 3.** Examples of cross sections for the light isotopes hydrogen, an efficient moderator, boron as an efficient absorber and silicon, which can be considered to be nearly transparent, from the ENDF library from thermal energies ranging to several MeV.

whereas for 'inelastic' cross sections only the main contributors are summed up, see Tab. A1, and 'absorption' itself is under-
stood as a sum of MT numbers stated in Tab. 1, which can either lead to capture without consecutive particles or the creation of new neutrons by for example evaporation or charged particle ejection by converters.

### 4.2 Geometry

URANOS uses analytical geometry definitions and voxels as introduced in sec. 3. The following top-down structure is applied for describing the simulation environment:

$$\text{geometry} \rightarrow \text{layer} \rightarrow \text{voxelmesh} \rightarrow \text{material} \rightarrow \text{isotope}$$

Each layer of the stack is either entirely composed of a material or subdivided into several sections using a two-dimensional matrix from which voxels are extruded. The entities are filled with predefined 'materials'. A material is a specific composition
of isotopes with atomic weight and density. Tab. 2 provides an example of such a definition, whereas all materials available in URANOS can be found in appendix B. Most compounds are taken from McConn Jr et al. (2011). The voxel mesh is



**Table 2.** Example composition of the material 'dry air' and a neutron converter.

| Material | Density | Composition |
|---|---|---|
| Air | $1.2\,\text{kg/m}^3$ NTP | $78\,\%\ ^{14}\text{N}_2$, $21\,\%\ ^{16}\text{O}_2$, $1\,\%\ ^{40}\text{Ar}$ |
| Boron | $2.46\,\text{g/cm}^3$ | $80.1\,\%\ ^{11}\text{B}$, $19.9\,\%\ ^{10}\text{B}$ |

automatically loaded and generated if a file with a name corresponding to a layer number is found. It can be either a tab-separated ASCII matrix of equal row and column rank or a quadratic portable network graphics (PNG) image. The integer values $w$ or grayscale values denote the material numbers which primarily override the global layer definition. Typically solids are directly extruded from these values, yet there are four further declaration modes:

- the material is soil and $w$ defines the amount of water in volume percent,

- the material is soil and $w$ defines the porosity,

- the material is defined globally by the layer or by voxels and $w$ scales the density,

- the material is defined globally by the layer, $w$ scales the height of this material and the remaining volume extended to the full layer height is filled with air to represent the remaining soil porosity.

The layers can be stacked on top of each other with individual definitions to realize complex geometries. Fig. 4 provides examples to illustrate the scope of applications and the scales which can be targeted. The images of one single layer act hereby as a sectional view. Especially landscapes can be modeled using the last declaration mode, an example is provided in Fig. 4. The geometry of each layer is simply defined by an array of eight elements:

$$g = [x\ \text{lowerbound}, x\ \text{upperbound}, \tag{21}$$

$$y\ \text{lowerbound}, y\ \text{upperbound},$$

$$\text{upper}\ z\ \text{position}, \text{height}, \text{material}, \text{layernumber}],$$

whereas the lateral lower and upper bounds are defined globally and the layer number acts as an additional identifier to create subgroups within the stack. Furthermore, the forward and backward propagation direction are defined according to if the layer number along the path increases or decreases, respectively.

Neutron tracks $\boldsymbol{S}$ are described by a mixed geometry definition of support vectors $\boldsymbol{x}$ in Cartesian coordinates and spherical direction vectors $\boldsymbol{r}$:

$$\boldsymbol{x} = \begin{pmatrix} x \\ y \\ z \end{pmatrix} \quad \text{and} \quad \boldsymbol{r} = \begin{pmatrix} r \\ \vartheta \\ \phi \end{pmatrix}, \tag{22}$$





**Figure 4.** Examples of layers for voxel geometry definitions (all in top view) with grayscale values defining preconfigured materials: a) a 2 " proportional counter with 1 " of moderator, b) the rooftop of the institute of physics in Heidelberg, c) a part of a lake (Schrön, 2017), d) voxel geometry for a digital environmental model (Kaunertal Glacier at N46° 52.2 E10° 42.6) with $500 \times 500$ pixels at a lateral resolution of 1 m and 0.5 m in height, e) shaded illustration of the resulting layered voxel structure from d).

denoting the three spatial coordinates $x, y, z$ and the angles $\vartheta, \phi$ with the range $r$. The choice for this system is due to the fact

305   that this characterization provides direct access to the necessary observables. Examples are point sources which are randomly distributed in both angles or detector planes for which the beam inclination is an important parameter considering sensitivity. Hence new coordinates $\boldsymbol{x}'$ are calculated by

$$
\boldsymbol{x}' = \begin{pmatrix} x \\ y \\ z \end{pmatrix} + \begin{pmatrix} r\cos(\phi)\sin(\vartheta) \\ r\sin(\phi)\sin(\vartheta) \\ r\cos(\vartheta) \end{pmatrix}
$$





and for determining the position on a layer at elevation $z_L$

$$\boldsymbol{x_L} = \begin{pmatrix} x \\ y \\ z \end{pmatrix} + (z - z_L) \begin{pmatrix} r\cos(\phi)\tan(\vartheta) \\ r\sin(\phi)\tan(\vartheta) \\ 1 \end{pmatrix}.$$

Time is an indirect quantity. It is derived from the geometrical position of the neutron calculated from energy and initial conditions.

In URANOS three layers can each be assigned specific functions. These are source layer, detector layer and ground layer. The source layer defines the origin for all neutron histories. Especially all height values for starting positions, see also sec. 5, are restricted to be initiated here. This layer may neither be the upper- nor lowermost as otherwise neutrons would escape the computational domain. The ground layer is used in cosmic neutron simulations to record the spectra at the air/ground interface and monitor the penetration depth. In the detector layer either single real or virtual detectors can be placed, or the layer itself acts as a virtual detector and records every neutron passing, see also sec. 7.

## 5 Sources and energy

URANOS provides a variety of sources. A source is defined by a spatial distribution and an energy spectrum from which random values are sampled. They are either defined as

- point sources with all neutrons starting from the same coordinate vector,

- a plane source with all neutrons sharing the same $z$ coordinate within lateral boundaries,

- a volume source, which randomly distributes neutrons in the source layer within lateral boundaries, and alternatively extends the volume source downwards to the ground layer with exponentially distributed height values.

As explained in sec. 4.2 the source has to be placed in the source layer, which defines its $z$-position. For the coordinates $(x, y) \in A$ in the source area $A$ in case of plane or volume sources the options are:

- rectangular boundaries with either equal aspect ratio (square) or any other, sampling the origins uniformly from possible positions in $(x, y)$, and

- circular boundaries, sampling the origins either uniformly in radius $r$ from the center or in $(x, y)$.

Furthermore, the starting angle $\vartheta$ can be set to either:

- full or half sphere, sampling $\vartheta$ in $[0 \dots \pi]$ or $[0 \dots \pi/2]$,

- unidirectional beam, which allows to set $\vartheta$ to a specific inclination. Additionally, a divergence $s_\vartheta$ can be chosen. Then, angles are sampled from a gaussian function centered around $\vartheta$ with a width of $s_\vartheta$.





The starting energies are derived from normalized distributions, which are described in the following sections. For source definitions on a linear support in $[a, b]$, like in sec. 5.2, the random variable $\xi \in [0, 1]$ is scaled to the abscissa test quantity

$$\xi_t = a + (b - a)\xi.$$

For source definitions on a logarithmic support in $[10^a, 10^b]$, like in sec. 5.1, $\xi$ is scaled to

$$\xi_t = 10^{a + (b - a)\xi}.$$

## 5.1 The cosmic neutron source

The cosmic neutron source definition is specifically designed for the problem of soil-moisture-dependent neutron transport in the vicinity of the atmosphere-soil interface. Instead of propagating primary particles through several kilometers of atmosphere, a source definition near the ground level is chosen. Recent works, especially from Sato and Niita (2006), Sato et al.
(2008) and later Sato (2015), have provided analytical functions modeling cosmic-ray spectra for various conditions like atmospheric depth and cutoff rigidity. In their latest version Sato (2016) introduced a 'black-hole' mode which allows to exclusively model the downward-oriented component of the flux, which is used as the default cosmic neutron spectrum. Fig. 5 shows the URANOS cosmic-ray neutron spectrum (black) and exemplarily the total spectrum above ground for 10 % soil moisture. The energy of neutrons can range over more than 12 orders of magnitude. The plot here as well as the following will be presented
logarithmically in units of lethargy. The intensity $I$ or flux density per logarithmic unit of energy is given in units of

$$I = \mathrm{d}\Phi/\mathrm{d}(\log(E)) = E\mathrm{d}\Phi/\mathrm{d}E. \tag{23}$$

## 5.2 General sources

Besides the cosmic neutron source definition, energy distributions for various specific sources have been implemented. These additional spectra can be used to investigate special applications with regard to sensor development, e.g. sensitivity to thermal
neutrons. These available source configurations allow sampling from thermal as well as fission spectra. Exemplarily some are shown in Fig. 6.

    – **Monoenergetic**: neutrons of energy $E$ or wavelength $\lambda$,

    – **Thermal**: neutrons at a temperature $T$ described by a Maxwellian distribution

$$N(E) = \frac{E}{(k_B T)^2} \exp\left(-\frac{E}{k_B T}\right). \tag{24}$$

– **Predefined**: americium-beryllium spectrum from ISO group 85/SC 2 Radiological protection (2001),

    – **Evaporation**: assuming the nucleus to form a degenerate Fermi gas (Weisskopf, 1937) one can derive various forms of density distributions in the form

$$N(E) \propto E \exp\left(-\frac{E}{k_B T}\right), \tag{25}$$





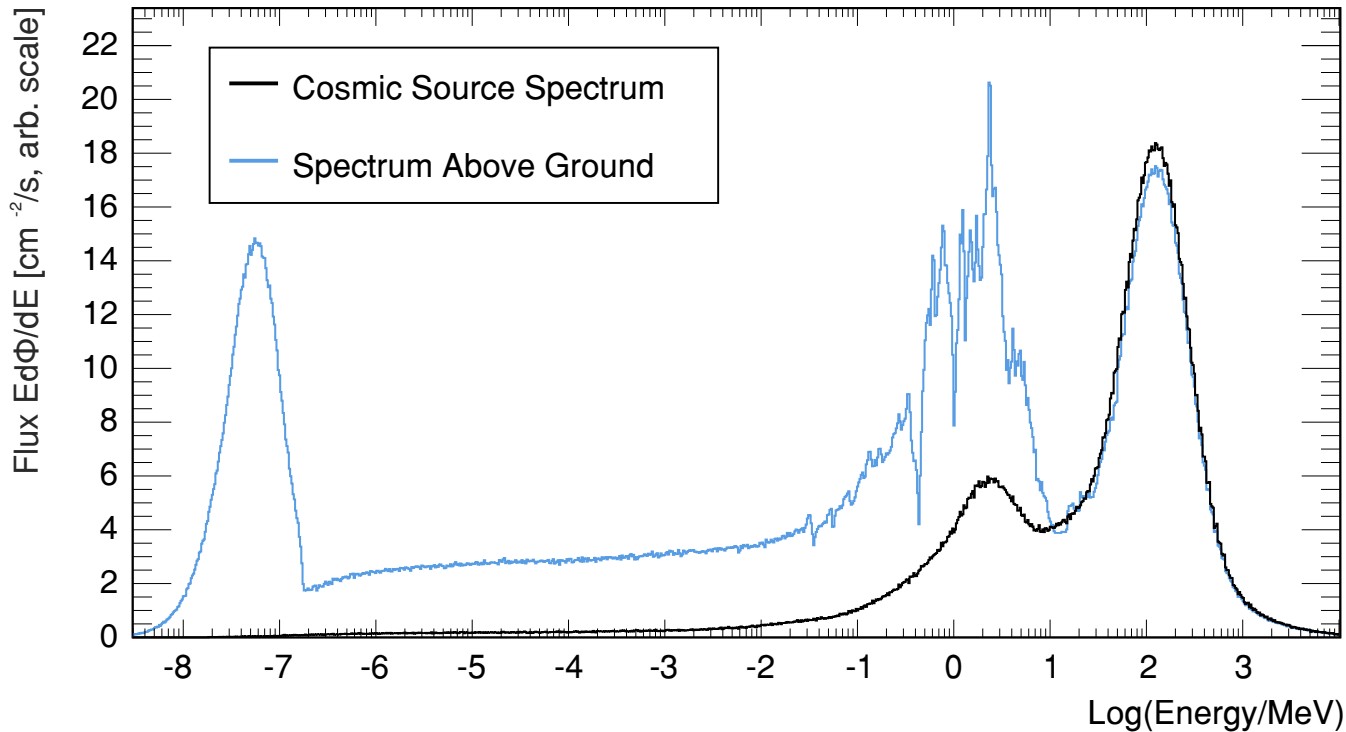

**Figure 5.** The URANOS Cosmic Neutron Source Spectrum (downward-only, black) and total angular integrated flux after interaction with the soil 50 m below the source (blue).

which are simply described by a temperature parameter (Terrell, 1959). The energy distribution of the neutrons released
by fission are commonly represented either by a Maxwellian distribution or the following Watt spectrum (Iyer and Ganguly, 1972).

– **Fission**: A semi-empirical description for fission neutrons is the Watt spectrum (Watt, 1952), especially used for $^{235}$U, which can be selected as a source although the isotope itself is not implemented. The following spectra can be selected

$$N(E) = 0.4865 \sinh\left(\sqrt{2E}\right) \exp(-E) \tag{26}$$

and for $^{252}$Cf (Smith et al., 1957)

$$N(E) = \sinh\left(\sqrt{2E}\right) \exp(-0.88E), \tag{27}$$

which are both specific cases of the more general form of a Maxwellian distribution. A more accurate modeling can be performed by specifying the Watt parameters $a$ and $b$ taking into account the mean neutron kinetic energy of and those of the fission fragments:

$$N(E) = \frac{\exp(-ab/4)}{0.5\sqrt{\pi a^3 b}} \sinh\left(\sqrt{bE}\right) \exp\left(-\frac{E}{a}\right). \tag{28}$$





The parameters $a, b$ are usually tabulated as a function of energy, element and isotope in ENDF libraries.

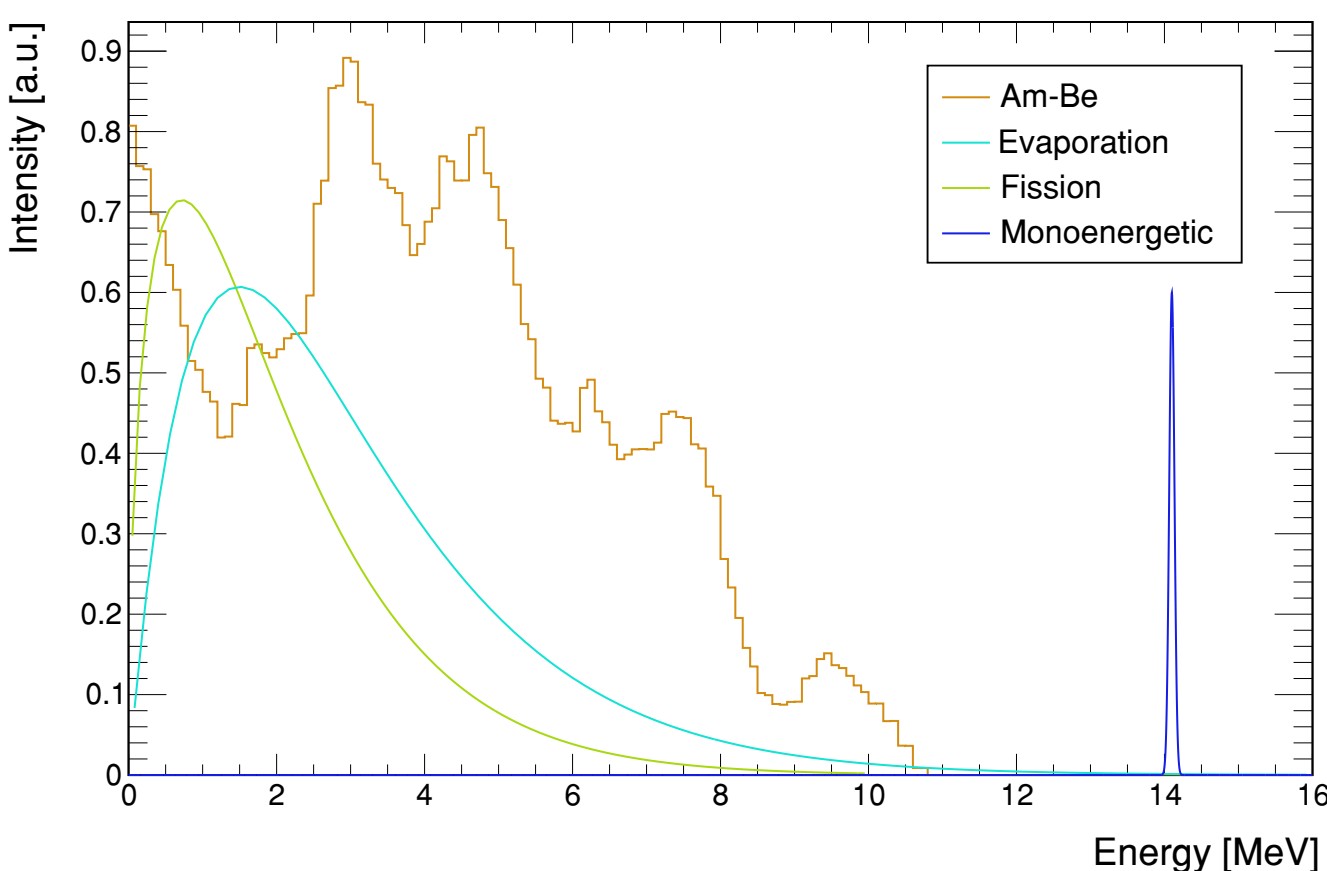

**Figure 6.** Different preconfigured source distribution functions in URANOS in the MeV range covering different use cases: laboratory test sources like americium-beryllium (yellow), spontaneous fission (green), evaporation (cyan) from nuclear de-excitation and fusion (blue)

## 6 Calculation scheme

### 6.1 Loop nodes

The main calculation routine runs in two loops, for each neutron in the neutron stack, the geometry stack is traversed layer-by-layer

$$\boxed{\text{Neutron Stack}} \longrightarrow \boxed{\text{Layer Stack}}$$

Each onset neutron is a placeholder and only initialized at runtime with the particle number not being conserved due to physical

processes generating further neutrons. The layer stack is created at startup and consists of a fixed amount of elements which are traversed by an iterator either forwards or backwards, depending on the spatial direction vector.





The possible initial conditions for neutrons are

- energy: available source definitions from sec. 5, which can be either real values, normalized functions to be sampled from or lookup tables.

- geometry: definition from sec. 4.2, which can be either a fixed vector from a source, a distribution function to be sampled from or lookup tables, which are normalized at startup.

Using these initial conditions the loop over the layer stack commences. Each layer, which is geometrically described in sec. 4.2, can either consist of a homogeneous material defined by its isotope composition, a material defined by an analytical function, or an input matrix from which voxels are extruded. A comprehensive material list is provided in appendix B. The neutron

iterates to the following layer if it geometrically leaves the boundaries without absorption and no change of materials can be found in the collision detection.

### 6.2 Tracking in finite geometry regions

For each layer the material composition is loaded according to the actual position of the neutron. The definition either accounts for the whole layer or for regions, which can be described by analytic functions or voxels. For the selected material the total

macroscopic cross section $\Sigma_t$ is set up isotope by isotope. The amount and type of reactions (MT identifiers), depends on the element, see also the description in sec. 4.1 and the isotope list in appendix A. Elemental hydrogen for example cannot undergo inelastic scattering and $^{10}$B exhibits a negligible radiative capture probability, so only charged reaction paths are relevant. The selection criteria in detail are

- elastic and absorption cross sections are always calculated if available.

- inelastic cross sections are loaded on demand for energies $750\,\mathrm{keV} < E < 50\,\mathrm{MeV}$.

The macroscopic cross section of a compound with weight fractions $w_i$ of $n$ elements and energy dependent absorption $\sigma^a$ and scattering $\sigma^s$ cross sections is defined as

$$\Sigma_t = \rho N_A \sum_{i=1}^{n} w_i \frac{\sigma_i}{M_i} \ \ \text{with } \sigma_i(E) = \sigma^a(E) + \sigma^s(E). \tag{29}$$

Other cross sections can also contribute. The free path length $l$ is sampled from a random number $\xi$ as described in sec. 2.3

from (5): $l = -\ln(\xi)/\Sigma_t$. In case the material definition contains a density multiplication factor, it is applied to $\Sigma_t$ before evaluating (5). The distance to the border $l_{\mathrm{trj}}$ is calculated by the $z$-coordinate of the last interaction of the neutron $z_0$ and the layer $z$-position $z_l$ and height $d_l$.

In case the material is defined by voxels, additionally a procedure is applied, which samples the trajectory according to the underlying pixel matrix:

- determination of the z-projected length $z_m$ of one lateral unit pixel $s_p$ for the actual direction vector by $z_m = s_p/\tan(\vartheta)$. The unit pixel size is determined by the spatial extension of the domain divided by the number of pixels.



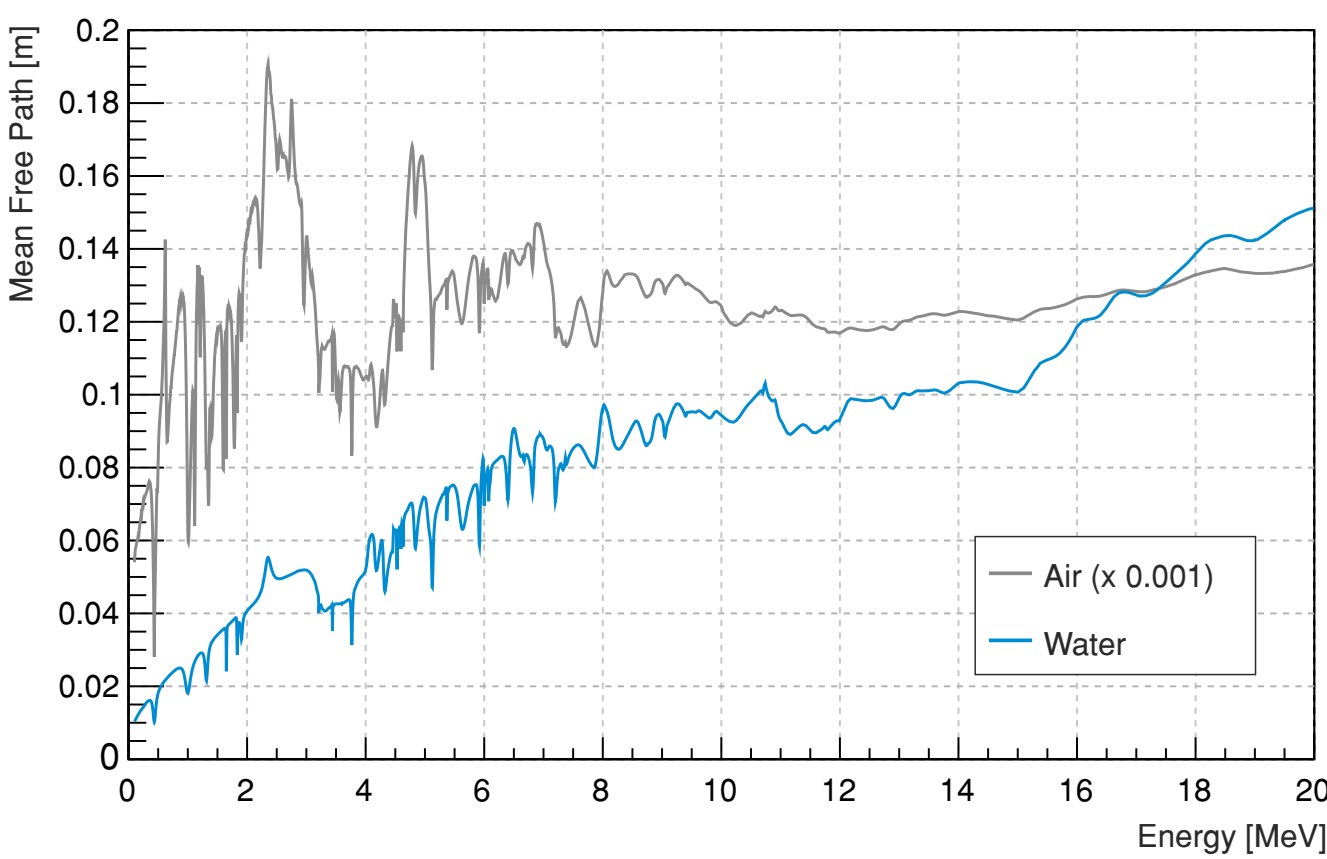

**Figure 7.** Mean free path $1/\Sigma_t$ for neutrons in the MeV range. The dominant peaks originate from the contribution of the elastic scattering cross section, in dry air (NTP) mainly by nitrogen, in water by oxygen, see also Fig. 8.

 – If the material of the voxel at $\boldsymbol{x}'$ for $z_0 \pm z_m$ is different from the actual, stop and repeat the range calculation for the actual composition and geometry. If the material does not change iterate $\pm z_m$ until the layer border is reached. The propagation direction, forward or backward, determines $\mathrm{sgn}(z_m)$.

If $l_{\mathrm{trj}} > l$ no interaction takes place and the neutron can proceed to the following layer. It has to be noted that for a voxel-based geometry definition the resolution is the size of a pixel. A neutron in that case will not interact with the side faces of a voxel but with its volume. If $l_{\mathrm{trj}} < l$ the spatial coordinates of the interaction $\boldsymbol{x_i}$ are calculated by

$$
\boldsymbol{x_i} = \begin{pmatrix} x_0 \\ y_0 \\ z_i \end{pmatrix} + \begin{pmatrix} \cos(\phi)|\tan(\vartheta)(z_i - z0)| \\ \sin(\phi)|\tan(\vartheta)(z_i - z0)| \\ 0 \end{pmatrix}, \tag{30}
$$

with afterwards updating the new $z$ coordinate.





Consecutively, the type of reaction is determined by another random number $\xi$. The relative fractions of the respective macroscopic cross section define the probability:

$$
\begin{aligned}
\xi < \frac{\sigma_{\text{el}}}{\sigma} &\longrightarrow \quad \text{scattered elastically,} \\
\frac{\sigma_{\text{el}}}{\sigma} < \xi < \frac{\sigma_{\text{el}} + \sigma_a}{\sigma} &\longrightarrow \quad \text{absorbed,} \\
\xi > \frac{\sigma_{\text{el}} + \sigma_a}{\sigma} &\longrightarrow \quad \text{scattered inelastically,}
\end{aligned}
\tag{31}
$$

as the probability of selecting a reaction type $i$ from all possible interaction channels is

$$
p_i = \frac{\Sigma_i}{\sum\limits_{j=0}^{n} \Sigma_j} = \frac{\Sigma_i}{\Sigma_t}.
\tag{32}
$$

The target interaction element is determined by a randomly choosing from a proportional lookup table. Each reaction type is accompanied by two `vectors` - one number represents the cumulative probability distribution $v_s$ and one number a list of corresponding elements $v_e$[1]:

$$
v_s[n] = \left\{ i \,\Big|\, \sum_{j=0}^{i} \Sigma_j \right\},
\tag{33}
$$

$$
v_e[n] = \{ i \,|\, \text{isotope of } i \}.
\tag{34}
$$

For inelastic scattering additionally the excited state of the target isotope has to be determined. One `vector` of numbers contains the cumulative cross section distribution and two support `vectors` contain the $q$-values representing the energy loss in MeV and the inelastic angular distributions in a `<TMatrixF>` list. The individual contributor, which will be used to

determine the reaction target, is chosen by a random number $\xi$. If

$$
v_s[i] \le \xi \le v_s[i+1], \; \forall i > 0,
\tag{35}
$$

then the corresponding isotope is taken from $v_e[i]$[2].

### 6.3  Interaction channels

For each interaction the following quantities are updated:

– the position vector $x$, including time, by adding the path length $l$ to the last position,

– the direction vector $r$,

– energy, including velocity $v$ and wavelength $\lambda$.

---

[1]Example: Natural boron, see also Tab. 2, contains $\approx 80\,\%$ $^{11}$B and $\approx 20\,\%$ $^{10}$B. For a reaction the cross section $\Sigma_i = \Sigma_i\left(^{10}\text{B}\right) + \Sigma_i\left(^{11}\text{B}\right)$ is accompanied by a `vector` of individual contributions $[\Sigma_i(0), \Sigma_i(0) + \Sigma_i(1)]$ and a `vector` of isotopes $[^{10}\text{B}, ^{11}\text{B}]$.

[2]If $i = 0$, then $\xi \le v_s[1]$.





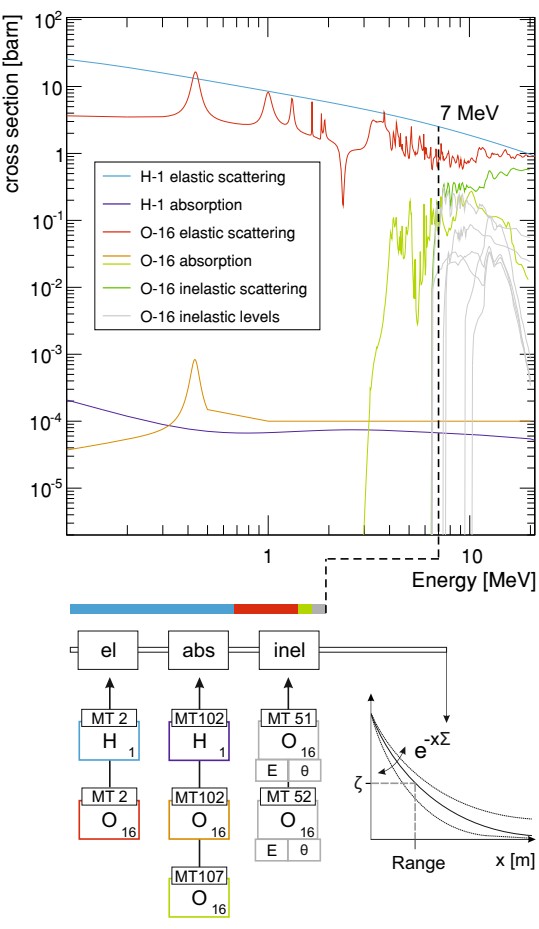

**Figure 8.** Range calculation in URANOS: For a given neutron energy, here in the MeV range, the cross sections from the isotope list are evaluated according to elastic, inelastic and absorption processes. Only possibly relevant contributions are evaluated. The left panel shows such a list of reaction probabilities for water. Inelastic levels are only displayed up to MT56 and are linked additionally to energy loss and angular distribution. The cross section multiplied by the atom number density (29) yields the macroscopic cross section. By sampling a random number $\xi$ as in (5), a free path length value for the range (4) is obtained.

### 6.3.1 Elastic and inelastic scattering

Scattering is described by the collision of a neutron with a nucleus of mass $A$ assuming energy and momentum conserva-

tion.The problem has a radial symmetry regarding the impact parameter, therefore only one angle $\vartheta_{\mathrm{CMS}}$ needs to be calculated. The second angle can be determined by a random number $\xi$ in $[0,1)$

$$\phi_{\mathrm{cm}} = \pi \left(2\xi - 1\right). \tag{36}$$





For inelastic scattering the energy loss is substituted by the $q$-value obtained from (35) and (34), respectively. The target velocity $V$ can be neglected for kinetic energies $E$ of the neutron:

$$V \approx 0 \text{ if } \begin{cases} 0.11\,\text{eV} < E < 1\,\text{MeV} & \text{in case of hydrogen,} \\ 0.15\,\text{eV} < E < 0.01\,\text{MeV} & \text{otherwise.} \end{cases}$$

For lower energies the interaction result has to be calculated by laws of thermal scattering taking into account the velocity distribution of the target material. In case of amorphous material or fluids there is no analytical form to describe such, therefore only sampling from an effective thermal spectrum like (24) is carried out. For solids with a crystal lattice Bragg scattering is the dominant channel. The kinetic theory of gases allows a cohesive description of the scattering process. For such the energy

and angle are sampled according to (17), (18) and (19).

In case of higher energies than stated in the above limits the angular distribution in the center of mass frame can be found in ENDF cards either tabulated or described by Legendre polynomials. With increasing energy the forward direction is preferred, except for hydrogen - here the asymmetry is much weaker than for heavy elements and only for very high energies a significant deviation from an even distribution can be observed.

For inelastic scattering with an energy transfer $E^*$ the evaluation of the angular distributions is carried out likewise, whereas the lowest energy, for which the reaction can occur, is given by the first $q$-value. Hence, the reaction kinematics of inelastic processes share some similarities with elastic processes of corresponding kinetic energies $E' = E - E^*$.

As the scattering kinematics have been calculated in the center of mass system, a transformation to the laboratory system is carried out via

$$\vartheta_l = \arccos\left( \frac{1 + A\cos(\vartheta_{\text{cm}})}{\sqrt{A^2 + 1 + 2A\cos(\vartheta_{\text{cm}})}} \right), \tag{37}$$

and added to the existing direction vectors

$$\vartheta_u = \cos(\vartheta^{\text{o}})\cos(\vartheta_{\text{cm}}) + \sin(\vartheta^{\text{o}})\sin(\vartheta_{\text{cm}})\cos(\pi + \phi_{\text{cm}}), \tag{38}$$

$$\vartheta_l^{\text{new}} = \arccos(\vartheta_u), \tag{39}$$

$$\phi^{\text{new}} = \phi^{\text{o}} \pm \arccos\left( \frac{\cos(\vartheta_{\text{cm}}) - \cos(\vartheta^{\text{o}})\vartheta_u}{\sin(\vartheta^{\text{o}})\sin(\vartheta_{\text{cm}})} \right). \tag{40}$$

Due to the choice of the coordinate system, see also the geometry definition (22), adding direction vectors is less convenient than the otherwise usual declaration. The method presented here equals to an Euler rotation in $\theta$ and $\phi$ around the direction axis given by the trajectory of the particle.

### 6.3.2 Evaporation

URANOS simplifies the calculation of the evaporation process as in the low-Z and intermediate energy range most quantities

relevant for fissionable elements are approximately constant. The mean number of evaporated neutrons can be considered constant $\overline{n}_{\text{evap}} \approx 1$ for projectile energies below several hundred MeV and mass numbers of $A < 100$ (Cugnon et al., 1997). Furthermore, for the emission energy a Maxwellian spectrum according to (25) with a mean neutron energy of 1.8 MeV (Kawano





et al., 2013) and a flat angular distribution (Bramblett and Bonner, 1960) is assumed. In order to provide upper limits in comparison: $^{235}$U produces on average $\approx 2.4 + E$/MeV neutrons per fission.

### 6.3.3 Absorption

Neutrons are either absorbed by a non-radiating process and consequently the calculation is terminated or the material is a specific absorber, which leads to a scoring by the detection unit. A specific case is the high-energy cascade: URANOS mainly carries out neutron interactions. For the generation of high energetic radiation in the atmosphere charged particles are also largely contributing to the production of the neutron component (Sato, 2016; Ziegler, 1998). As far as for low energetic and albedo neutrons such can be neglected, in order to simulate more than 100 m of atmosphere the generation of the primary spectrum is emulated by an effective model: For any interaction occurring above 16 MeV with the possible release of secondary neutrons the primary neutron is not eliminated if a random number $\xi$ is below a specific value $k_{\mathrm{HE}}$, receiving only a fractional energy loss and angular deviation. This value $k_{\mathrm{HE}}$ is tuned to emulate an effective atmospheric attenuation length $L_{\mathrm{prim}}$ of the primary spectrum component of 145 cm$^2$/g. Experimental values for $L_{\mathrm{prim}}$ are in the range of (135–155) cm$^2$/g, depending on the latitude of the site. Here, the value from Sato (2016) is taken.

## 7 Detector configurations

Neutrons can be scored in three different ways: One layer can be defined the 'detector layer', a virtual entity which can record any particle of chosen characteristics to pass through. Secondly, a virtual detector with limited spatial extension can be placed within the detector layer. Specific materials, if voxel definitions are used, can additionally score neutrons like the virtual detector. For a thermal neutron detector this material would be the converter. Output options are either only hits, partial tracks within a material or full tracks. The detector layer stretches out to the full domain dimension and is placed at a fixed height. It is most useful for mapping the spatial distribution of the neutron flux. The virtual detector can emulate an instrument at a specific position and can be used for analyzing the origin distribution of neutrons. It can be set to transparent for a generalized recording or to absorbing for mimicking a detector more realistically.

**Scoring options for CRNS**

In the most simple case a uniform detection efficiency $\epsilon$ can be chosen within a specific range

$$\epsilon = \begin{cases} 1 & \text{for } E_{\min} < E < E_{\max}, \\ 0 & \text{otherwise.} \end{cases} \tag{41}$$

As far as thermal neutrons are not considered, the flux in the epithermal/fast region, can be considered a plateau region, partially justifying the established choice of (41). In order to not model an entire cosmic-ray neutron detection system in an environment larger by orders of magnitude, a set of 'standard' detectors has been simulated independently and integrated as an effective model. In many possible scenarios the 'simple' scoring option (41) leads to incorrect results. This approach therefore allows to





directly analyze the neutron flux measured by an actual device and at the same time allowing to scale up the scoring volume beyond the physical dimensions of the instrument. Fig. 9 shows the implemented functions, which represent averaged values for different side faces of a detector. In URANOS cubic spline interpolation is used for describing the absolute efficiency and

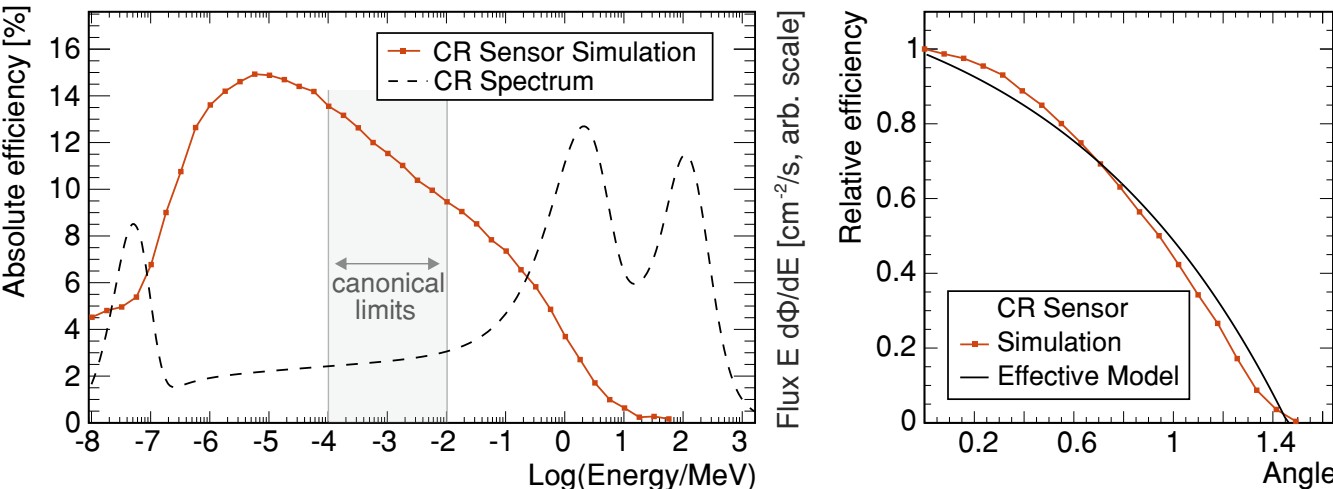

**Figure 9.** . Detection efficiencies for the 'standard' CRNS detector model. (left) Energy dependent absorption probability for perpendicular irradiation, here: simulation of a monoenergetic beam with results (red markers) averaged over the surface. (right) Energy independent, averaged, angular dependence relative to the left panel.

the angular dependence is modeled by

$$\epsilon_\vartheta = 1.24 - 0.254 \exp\left(\frac{\theta}{0.92}\right). \tag{42}$$

The options above can be applied to the whole detector layer. If for example angular resolution or single neutron tracking information is required one can place two types of scoring units within the detector layer, which is either a plane in $z$-direction, a sphere or a vertical cylinder, whereas in both cases the radius can be specified. The cylinder height corresponds to the detector layer. If due to positioning or the choice of the radius of the sphere an intersection with the layer boundary occurs, only the volume inside the detector layer is taken into account.

## 8 Evaluation of the URANOS model

As the implemented physics is similar, model results are expected to be comparable to other codes. In preparatory studies, we explored the performance of the URANOS model in reproducing results from standard software like MCNP(X). The tests successfully agreed in many different setups as the one presented by Köhli et al. (2021). However, more rigorous validation experiments may be conducted in future projects using MCNP and GEANT4. Apart from that, URANOS model results have found remarkable agreement with observations, e.g. in terms of footprint experiments (Köhli et al., 2015), and applications to





many studies in cosmic-ray neutron sensing (Heidbüchel et al., 2016). Furthermore, the physics of neutron attenuation in water can be validated with measurements from Caswell et al. (1957), conducting measurements in water tanks. Water is one of the main actors in neutron physics and the description of neutron moderation in URANOS performs adequately, as Fig. 11 shows.

## 8.1   Basic performance examples

In order to visualize the tracking capabilities of URANOS Fig. 10 shows two non-trivial neutrons paths from generation until absorption, exemplarily in air (top) and in the ground (bottom). It acts as a demonstrator for the interactions at this specific interface. In air the main scattering partners are nitrogen and oxygen, which leads to a large amount of scatterings with small energy decrements. By the long path lengths in the less dense medium the neutron also can acquire hundreds of meters of integrated travel distance. Inside the soil typical scattering lengths are far below one meter. For high energy neutrons, the main scattering partners can be silicon, aluminum and oxygen. However, due to the presence of water a few interactions with light nuclei can thermalize a neutron (blue lines). Then it will carry out a random walk which will be dominated by hydrogen scattering.

## 8.2   Diffusion length in water

The attenuation of fast neutrons by efficient moderators is a basic example of neutron physics and the main source of thermal neutrons. Modeling the slowing down process properly requires the correct description of interaction lengths, energy loss and geometric transport. Therefore, it can be regarded as validation test of the Monte Carlo code. Only few examples of well controlled and simple measurements can be found in available literature. Caswell et al. (1957) describe an experiment of determining the radial distribution of neutrons in a water tank from 14.1 MeV to thermal energies and 1.46 eV. A deuterium beam is delivered by an aluminum tube onto a tritium target inducing fusion. The tank measures 2.4 m in length and 1.2 m in height, whereas the particle injector is located at a distance of 0.6 m from one wall and vertically centered. The flux is measured pointwise by indium foil activation, which provides data for the non-equilibrium state above 1 eV, and thermal neutron detectors with cadmium shielding. Although both energy regimes are supposed to exhibit similar range distributions they have to be treated by different methods of neutron transport. Until reaching the indium resonance a maximum mean energy loss by elastic collisions, including a few inelastic reactions, can be attributed to hydrogen interactions. Below this limit the kinetics of neutrons are dominated by thermal scattering leading to a constant average energy. Fig. 11 shows the measured fluxes from Caswell et al. (1957) in comparison to the simulation results. Both attenuation distributions are in good agreement. The particle density in both cases peaks at around 15 cm followed by a nearly exponential decay with similar attenuation lengths.

## 8.3   Simulation of the response function of Bonner Spheres

A similar case are Bonner Spheres (Bramblett et al., 1960), proportional counters surrounded by shells of polyethylene. As this spectrometer type of array is used to monitor environmental fluxes, various studies were carried out for the modeling of





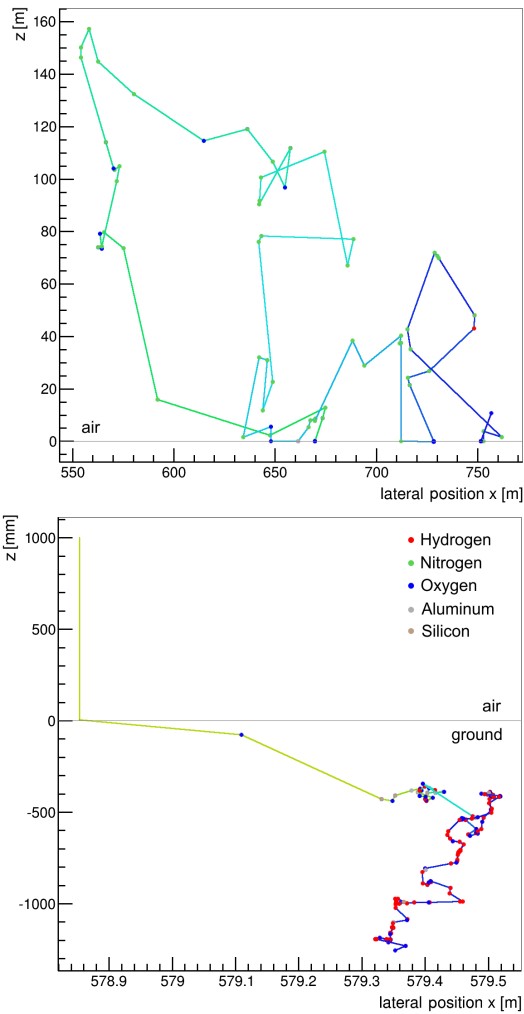

**Figure 10.** Projection of track calculations in an air-ground interface. The simulated neutrons, which are artificially released from 1 m above the soil, are rainbow-colored according to the logarithm of the corresponding energy scaling from 10 MeV (red) to thermal (blue). Top: a neutron which mainly scatters in the air. Bottom: a neutron thermalizing inside the soil. To be noted: both x- and y-axes are not scaled equally.

such (Hertel and Davidson, 1985; Mares and Schraube, 1994; Garny et al., 2009; Decker et al., 2015). Whereas the neutron
550  range distribution in water in the previous example demonstrated geometric transport and collision treatment, the Bonner
Sphere offers the possibility to focus on an energy-dependent comparison and on the interplay of moderator and absorber.
Among the various existing technical realizations the helium-based version was chosen, equipped with a 3.2 cm spherical
counter. For reasons of convenience, the whole model has been discretized in 17 layers, which are symmetrically arranged
around the center. Laterally the resolution by the pixel matrix was set to 1 mm, therefore the voxel size of a $X$ inch sphere is



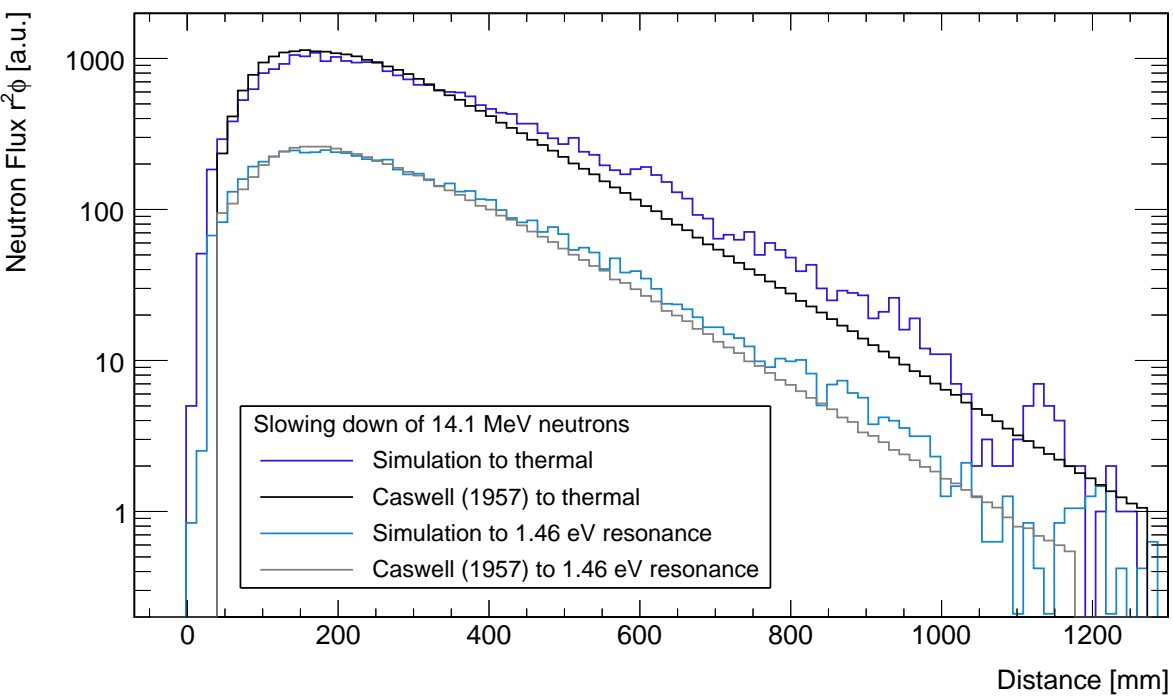

**Figure 11.** Comparison of the attenuation length from (Caswell et al., 1957) for deuterium-tritium fusion neutrons emitted into water. The spherical surface flux for thermal and indium resonance neutrons as a function of distance from the source is compared to the simulation results from URANOS.

1 mm × 1 mm × $(X/17)$ ". For the simulation the model was irradiated by a neutron beam of the same diameter as the sphere under an angle of $0\,^{\circ}$.

Furthermore, hydrogen atoms in polyethylene have been emulated by the scattering kernel derived from the (oxygen-) bound cross section in water. This can yield exclusively for thermal energies a systematic uncertainty of around 10 %. Due to the statistical nature of neutron transport the actual geometrical shape of a body has a minor influence compared to other parameters like overall volume or thickness. Exemplarily for the calculation routine some track views are shown in Fig. 12 as a central cut through the model and for the whole domain. Comparing to calculations from Mares et al. (1991), see Fig. 13, there is a good agreement in the energy sensitivity between response curves from literature and URANOS results. This successfully validates the simulation for basic scattering calculations.

## 8.4 Cosmic spectrum evaluation

Although since more than 50 years the general shape and height-dependent scaling of the cosmic-ray neutron spectrum at ground level is known (Yamashita et al., 1966), there is a perpetual discussion about precise features of the intensity distribution, especially at the soil interface. The reasons are that high-energy neutron interaction cross sections above 20 MeV







**Figure 12.** Flux calculation of Bonner Spheres of 2 inch diameter. The simulated neutron tracks ($E_{\mathrm{kin}} = 10\,\mathrm{keV}$) of $10^6$ histories are displayed in a central cross section of 3 mm height (top row) and the full domain of $13\,\mathrm{cm} \times 13\,\mathrm{cm} \times 5.4\,\mathrm{cm}$ (bottom row).

were originally not seriously investigated nor integrated into transport codes. Their evaluation and corresponding measurements are recent developments, mainly of the $21^{\mathrm{st}}$ century. Furthermore, the invention of the Bonner Sphere could standardize

dosimetric flux evaluations, yet, the neutron spectrum is measured indirectly. The flux distributions are result of unfolding algorithms (Sweezy et al., 2002), which rely only on a few absolute values and energy-dependent response functions from Monte-Carlo models of the detectors (Thomas and Alevra, 2002). This means that different simulations can produce slightly different weightings for different parts of the spectrum. In Fig. 14 an overview of different results from the most widely used codes is presented along with experimental results. The main differences appear in the high energy regime. These uncertainties

were partly compensated by effective nuclear interaction models. The peak structures at around 1 MeV, which are spectral lines of (in)elastic resonances, mostly oxygen, cannot be resolved experimentally by spectrometers and are displayed only at times.

    In order minimize this general problem URANOS uses a validated neutron spectrum near the surface as a source and releases it directly onto the ground to minimize typical uncertainties of atmospheric propagation. The implementation of the



**Figure 13.** Comparison of simulations of the energy dependent response function of Bonner Spheres of URANOS and MCNP calculations by Mares et al. (1991). The detectors are HDPE spheres with diameters in the range of (2–5) inch, equipped with a 3.2 cm $^{3}$He counter.

works presented by Sato (2016), which are based on Iwase et al. (2002) and Sato et al. (2008), are discussed in sec. 5.1. Fig. 15

presents the result from URANOS for the calculated neutron flux (green) above the surface in an infinite domain. It has to be pointed out that the resulting spectrum requires nearly the full physics and tracking computation.

On the qualitative level the underlying physics model correctly calculates the response to the soil. Three peaks can be observed - the high energy domain around 100 MeV with the incoming-only flux, the region around 1 MeV with the evaporation peak and the thermal peak at 25 meV. As there are no significant sources in the range of 1 eV to 0.1 MeV, there is a flat plateau

between neutron generation and thermalization. This plateau can feature a slant angle in case there are significant absorption processes involved. The incision between high energy and evaporation peak originates from not using cascades but an effective



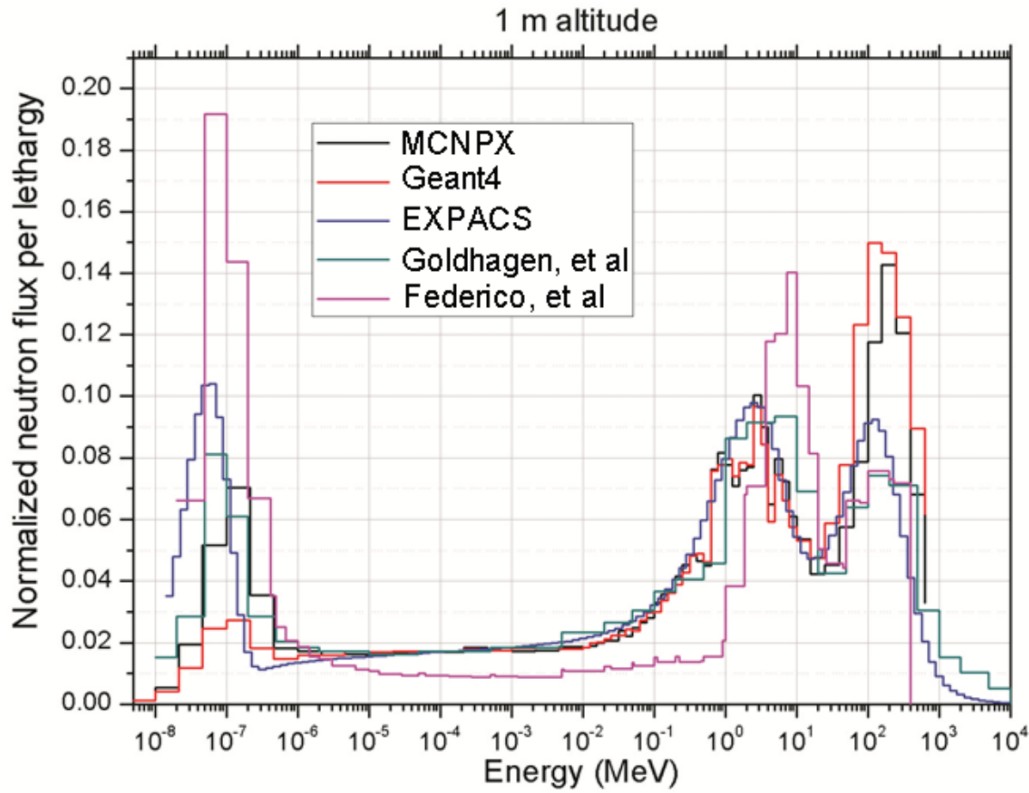

**Figure 14.** Energy dependent neutron flux at 1 m altitude calculated by EXPACS, MCNPX, GEANT4 and determined experimentally from Goldhagen et al. (2002) and from Federico et al. (2010). Environmental conditions are not the same. (Pazianotto et al., 2015)

model for neutrons above 20 MeV. For a quantitative investigation due to the lack of a generally accepted standardized spectrum or a consensus in the literature, the evaluation of the URANOS code focuses on the capability to reproduce the above-ground cosmic neutron spectrum for typical conditions. This implies that the input spectrum released on the ground should reproduce the same densities as the input formulae (blue) or other simulation toolkits like MCNP (black).

**8.5   Previous studies using URANOS**

The URANOS model has been in active development since 2014, while it has been employed by many international studies to support the CRNS data analysis and interpretation. We here provide a short summary of the published studies which indirectly confirm the proper modeling capabilities of URANOS by their comparisons to real-world field observations.

In the first application, Köhli et al. (2015) simulated the radial CRNS footprint and confirmed the results with measurements at the shoreline between land and a lake. Köhli et al. (2016) used an URANOS model of the Boron-10 based CASCADE detector including signal generation in the gas and its projection onto the readout structure in order to understand the signals generated specifically during Spin Echo measurements. The results agreed with reference efficiency measurements and helped





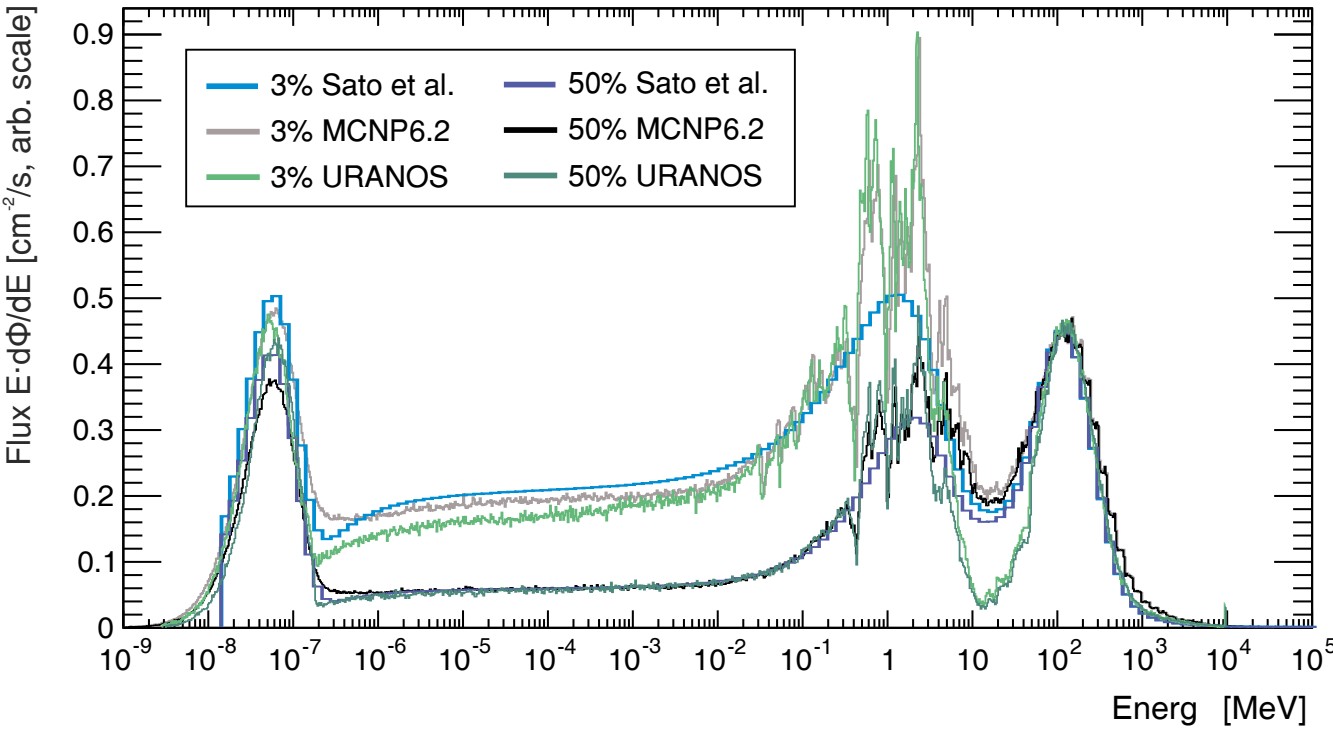

**Figure 15.** Cosmic-ray neutron spectrum comparison for 3 % (light colors) and 50 % (dark colors) soil moisture. The spectrum, which is generated from the analytical functions from Sato (2016), is shown in blue. The black-hole mode spectra from Sato (2016), see also Fig. 5, are used as input distributions at a height of 500 m and released onto the soil. The resulting intensity distribution is shown for MCNP (black) and URANOS (green) in comparison.

to improve the event selection algorithm. The model was also employed by Schrön (2017) to find the optimal location of a
buoy detector on a lake with minimal influence from the land. By challenging six different CRNS sites, Schrön et al. (2017) consistently improved their soil moisture measurement and calibration performance with the URANOS predictions of distance and depth contribution. This weighting approach has been later confirmed by many other authors worldwide. Schrön et al. (2018) simulated the spatial neutron density in an urban environment and confirmed agreement to corresponding observations. In Köhli et al. (2018a) URANOS was employed to generate reference track and energy deposition models in order to finetune
the event reconstruction in the BODELAIRE detector with its Timepix readout. Schrön et al. (2018) used URANOS to describe the local road effect for mobile CRNS applications and were able to experimentally verify and to correct these effects using the predicted relationships. Köhli et al. (2018b) simulated detector response functions of various neutron detectors used in the context of cosmic-ray neutron sensing and found very good agreement with previously published data and models. Li et al. (2019) were able to explain the observations in an irrigated orchard with the help of detailed URANOS simulations. Schattan et al.
(2019) discovered the hysteresis effect in CRNS signals of complex snow patterns and explained this effect accurately with dedicated URANOS simulations. Weimar et al. (2020) employed URANOS to develop new CRNS detector and shielding mod-





els which eventually improved the sensor performance. Köhli et al. (2021) revised the neutron-water relationship $N(\theta, h)$ and found substantially improved performance for CRNS observations particularly in dry regions. Badiee et al. (2021) employed URANOS simulations to support new developments of downward-shielded CRNS robots. Jakobi et al. (2021) investigated and
evaluated the footprint characteristics of thermal neutron transport using URANOS. Francke et al. (2021) used URANOS to explain the angular distribution of measurements with a directional-sensing CRNS variant. Rasche et al. (2021) were able to explain unconventional behaviour of neutron transport at a heterogeneous wetland site with the help of URANOS. The model also supported theoretical considerations by Schrön et al. (2021) about the influence of train wagons, rail tracks, and shielding material on a CRNS detector in moving trains, which then have been confirmed by experiments. Köhli and Schmoldt (2021)
investigated the possibilities of detecting unexploded ordnance using the pulsed neutron-neutron technique with the operation mode 'URANOS Downhole'.

## 9   Performance benchmarks

The performance of the code heavily depends on the simulation setup. The most significant contributor is the mean lifetime of neutrons in terms of scatterings within the domain. In simple configurations like for the analysis of detectors most neutrons
undergo only a few interactions before either being absorbed or leaving the domain. Atmospheric neutrons can have up to hundreds of scatterings before ending up thermalized. Additionally, the performance depends on the chosen materials and energy range, as for elastic scattering or absorption only a few cross sections like are evaluated. In the MeV domain, however, adding up all inelastic channels scales up to dozens of address requests. In order to provide practical estimations a standard setup can be defined as in Tab. 3. The domain measures $900\,\text{m} \times 900\,\text{m}$ with a source dimension of $840\,\text{m} \times 840\,\text{m}$. It contains
the minimum configuration of six layers.

**Table 3.** The standard setup for a layer composition in cosmic neutron sensing.

| Layer | Position | Height | Material | Function |
|---|---|---|---|---|
| 1 | $-1000.0\,\text{m}$ | $920.0\,\text{m}$ | air | top buffer layer |
| 2 | $-80.0\,\text{m}$ | $30.0\,\text{m}$ | air | source layer |
| 3 | $-50.0\,\text{m}$ | $47.5\,\text{m}$ | air | - |
| 4 | $-2.5\,\text{m}$ | $0.5\,\text{m}$ | air | detector layer |
| 5 | $-2.0\,\text{m}$ | $2.0\,\text{m}$ | air | - |
| 6 | $-0.0\,\text{m}$ | $3.0\,\text{m}$ | soil | ground layer |

A similar setup has also been used for simulating the so-called 'UFZ site' in Schrön et al. (2018) - an urban environment with many concrete buildings, streets, green spaces, railroad lines, a lake and trees. From the twelve layers in total eight contained a matrix of $1800 \times 1800$ pixels.

In order to provide references for the computational speed in realistic scenarios the following Tab. 4 summarizes the single
core performance of the code in terms of neutrons per second per GHz per core. The 'standard setup' and 'UFZ site' are





| Nº | name | description | Performance [n/(s·GHz·core)] | | |
| --- | --- | --- | --- | --- | --- |
| | | | URANOS | MCNP6 | GEANT4 |
| 1 | std. setup | water body, 5 g/m³ air humidity NTP | 930 | 300 | 70 |
| 2 | std. setup | ground with 10 % soil moisture, 5 g/m³ air humidity NTP | 450 | | 31 |
| 3 | std. setup | ground with 1 % soil moisture, 5 g/m³ air humidity NTP | 265 | 250 | 17 |
| 4 | std. setup | like Nº 1, with full domain tracking enabled | 710 | | |
| 5 | std. setup | like Nº 1, with thermal transport enabled | 260 | 260 | 16 |
| 6 | std. setup | like Nº 3, with thermal transport enabled | 130 | 220 | 9 |
| 7 | std. setup | like Nº 3, with thermal transport and full domain tracking enabled | 120 | | |
| 8 | UFZ site | with 10 % soil moisture | 500 | | |
| 9 | UFZ site | like Nº 8, without voxel geometry but same layering | 420 | | |
| 10 | detector | thermal spectrum onto a side face with $\vartheta = 0°$ | 9170 | | |
| 11 | detector | like Nº 10, with an americium-beryllium spectrum | 4060 | | |

**Table 4.** Benchmark results for the single core performance of URANOS (v0.99ρ) and MCNP for a number of practically relevant scenarios. URANOS was tested on a 4 GHz i7-6700K Skylake CPU in Windows 10 environment, MCNP6 on a 2.7 GHz E5-2680 Sandy Bridge CPU in Windows Server 2016 and GEANT4 (10.7) on a 3.4 GHz E-2124G Coffee Lake CPU in Debian 10.

described above and are simulated in combination with the cosmic neutron spectrum like presented in Fig. 15, the 'detector' is a rover-type instrument (Köhli et al., 2018b), which is a setup similar to the Bonner Sphere models and the other benchmarks are synthetic. Without additional voxel geometry descriptions URANOS requires approximately 230 MB of memory, mainly for storing ENDF data. The benchmark results show, that the performance of URANOS scales proportionally to the amount

of underlying calculations, i.e. scatterings. Applying a voxel geometry definition instead of a homogeneous layer has only a minor influence on the computational speed as long as the whole model can be kept in the system memory. Voxel models do not require collision tests for every surface within the domain, the number of intersection tests scales only by the voxel density. Compared to URANOS MCNP6 is less susceptible to the environmental topology or soil water content. From dry to moist conditions there is roughly a difference of 15 % and another 15 % can be gained by deactivating thermal neutron transport.

**10 User interface**

**10.1 Creation of input files**

The layered geometry of URANOS allows for building the geometry as a stack of materials. Especially some layers can be described by a single homogeneous material and others can contain complex patters constructed on the basis of images or ASCII matrices. This allows to easily realize for example a model domain composed of a soil with structures (e.g. buildings)

erected on top. The procedure is explained in sec. 4.2 and some examples for input image files are shown in Fig. 4. Beyond





the material input definition each layer can contain a density multiplicator matrix, which scales the density of the respective voxel by a certain factor and a porosity matrix which allows for adapting the soil bulk density. Images or matrices are stored in the active working folder and have to be labeled as the layer they refer to, for example `6.dat` or `6.png` define the geometry in layer 6. `6d.dat` would be the density matrix and `6p.dat` the porosity matrix. In order to create input image files, it is recommended to use a software which works on a pixel level. Contrasts must not be smeared out as usually done in photography software. Grayscale values between two colors represent completely different materials.

### 10.2 Command-line

URANOS can be run with different startup parameters. One of the options is to run the simulation without the GUI using a config file. This enables to create a batch file to start several jobs at once. The arguments that can be used in combination with `UranosGUI.exe` are the following:

– `silent`: disables console output

– `noGUI location/of/uranos.cfg`: starts the simulation without the GUI and takes all configuration options necessary for the model run from the `uranos.cfg`. The file also contains input and output directories. If no file is specified the configuration from the active directory is taken.

– `batchrun i j`: conducts a batch run with 11×11 = 121 values for soil moisture (1, 3, 6, 10, 12, 15, 20, 25, 30, 40, 50) in volumetric percent and air humidity (1, 2, 4, 7, 10, 12, 15, 18, 21, 27, 35) in g/m$^3$. These values are applied to the materials air and soil. `i` and `j` specify numbers for the setting to start and end with.

– `detectorBatchRun`: conducts a batchrun with 40 different energies from 0 eV to 20 MeV released orthogonally onto the domain ($\vartheta = 0$). The domain is supposed contain a detector model. An additional output file is created for the number of neutrons which are absorbed by helium-3 and boron-10.

### 10.3 GUI

The design of the graphical user interface provides all settings for adjusting the simulation alongside with the data output and information which are relevant to ensure a proper configuration at runtime. The main window is displayed in figures 16 and 17. The control bar (1) with its functions to start and stop the model run shows the simulation progress and the expected time to complete the simulation. The main user area is split into the settings pages (2) and the live view tabs (3). The sliders (4) control general features of air and soil, which can be used as materials in the geometry stack (5). After starting the simulation the neutron distribution within the domain can be viewed in the visualization tab (6). The cosmic-ray neutron spectrum above the ground provides a quick overview of the scattering and absorption characteristics of the domain.

In the left-side settings pages the available tabs contain a variety of configuration options, such as:

– Physical parameters: In the main control tab the central table is used to build the geometry stack by defining the height and thickness as well as the base material for each layer. Matrix definitions for materials, densities and porosity are



**Figure 16.** URANOS main user interface with: (1) main simulation control bar, (2) configuration tabs, (3) live-view tabs, (4) global environmental settings, (5) geometry stack with two layers defined by voxels, (6) birds-eye view of the neutron flux within the domain in the detector layer and (7) cosmic-ray neutron spectrum above the ground.

automatically loaded and assigned if the respective option is used. The sliders on the left side control the general settings for the moisture content of the materials air and soil. For the incoming spectrum the vertical cutoff rigidity can be chosen.


– Computational parameters: The lateral size of the domain and source can be adjusted in this tab alongside with specific options like boundary conditions or the activation of calculation models like thermal neutron transport. These options largely influence the computational speed and therefore need to be adjusted to fit the research question.

– Detector: In URANOS the settings for the detector and detector layer can be chosen independently. For both entities a flat detection efficiency within upper and lower energy bounds can be chosen or the 'physics model', which emulates the response function of an actual CRNS instrument. By using an enlarged virtual detector the simulation can gather

much more statistics than using a detector model with its actual dimensions as this significantly enlarged representation





can score neutrons using the same characteristics as a realistic model. Users can create own response functions and load them into the simulation.

– Showcase: The cosmic-ray input spectrum as well as the footprint can be viewed using this tab.

– Folders: This tab allows to set input and output folders for URANOS.

– Export: Which type of individual data sets to be exported from the detector and detector layer can be selected.

– Display: This tab contains the controls for the 'Live View' graphs, energy range, color scheme and scaling can be chosen. For the 'Birds-eye View' either hit or track distribution are available.

**Figure 17.** URANOS user interface showing the detector tabs. (left) Settings for changing energy and and angular sensitivity as well as geometric shape and extension. (right) Detector scoring output showing the origins of detected neutrons

The right-side visualization elements are:

– Birds-eye View & Spectra: The main tab shows the intersections of neutrons with the detector layer as a representation 700 of track or hit density. The whole layer can act as a virtual detector by applying the respective response function. The '+'





button opens a window with an enlarged high-resolution output of the currently shown detector layer display. The lower graph shows the energy distribution of neutrons above the ground. The full spectrum can be divided into an incoming part, which has been used as the initial spectrum and an albedo part which represents neutrons having at least one soil contact.

– Range View: In this tab the range distributions for detector and detector layer are presented which are also called horizontal weighting functions. The range is defined as the distance from the first collision in the soil to detection.

    – Spatial View: The top graph shows a horizontal cut view through the innermost 8 % of the detector layer for the chosen energy range. Furthermore in the depth distribution three plots are shown: The interaction of all neutrons within the soil, the maximum probe depth, which is the lowest point a neutron reached when being registered by the detector layer, and

the last probe depth, which shows the depth distribution of interaction points before leaving the soil and being detected. Both distributions can be regarded as approximations for the minimum and maximum depths for probing the soil.

    – Detector: This tab shows the points of first soil contact for neutrons scored by the virtual detector. By setting the 'range of interest' in the 'Display' tab to a value different from 'none' the plot changes from showing hits to the spatial density of origins.

## 10.4    Output formats

URANOS provides three output channels for the simulation data. The neutron flux distribution within the detector layer can be exported as an ASCII matrix or image. This data does only provide spatial information. The virtual detector can export the full information of neutrons passing through it: direction, energy, last interaction, first soil contact or model-dependent detection probabilities. Additionally, the full history, i.e. the complete neutron track list, can be exported. This feature allows

to back-trace only those neutrons which match specific detection criteria. All simulation data are furthermore stored in two compressed ROOT-files, one contains all graphs and distributions seen in the GUI itself and one contains support information mainly addressing mechanisms inside the domain like element- and materialwise scattering and absorption distributions.

## 11    Conclusions

URANOS is a new neutron Monte Carlo tool based on C++ with a graphical user interface specifically adapted to the needs of

environmental physics. It uniquely features a modeling geometry using the concept of voxels, three-dimensional pixels, which can be stacked in layers and extruded from grayscale images or ASCII matrices. Neither a 3D modeling software for vector objects is necessary nor the writing of elaborate steering files in order to realize complex landscapes and domain structures. A built-in, validated cosmic-ray neutron spectrum leads to quick solutions as it makes the extensive particle shower generation redundant. It allows to record neutron flux by a domain-wide scoring layer and a virtual detector with preconfigured instru-

ment characteristics. These built-in tools shorten the data analysis for modeling results without requiring programming skills. URANOS is now available for all users and regularly maintained to address the growing needs of the community.





*Code availability.* The URANOS source code is made available at the Github repository https://github.com/mkoehli/uranos/tree/URANOS.
URANOS v1.0 has been released under DOI: 10.5281/zenodo.6578668.
URANOS v1.0 is linked against ROOT 6.22.08, QT 5.15 and QCustomPlot 2.1.0.

*Data availability.* Libraries and data used in this publication have been released in the Github repository https://github.com/mkoehli/uranos/
tree/URANOS at the initial commit of URANOS v1.0.

*Sample availability.* Examples used in this publication and their configuration files have been released in the Github repository https://github.
com/mkoehli/uranos/tree/URANOS at the initial commit of URANOS v1.0.

## Appendix A: Elements, isotopes and reaction types

The database of URANOS materials relies on a library of predefined elements. Such are described by ENDF cards, which are extracted from the existing sources mentioned in section 2.5 and stored individually. The following Table A1 is a comprehensive list of isotopes, which have been selected and implemented.

## Appendix B: Material codepages

URANOS provides a list of already predefined materials, which are combinations of elements described in section A. Tab. B1
summarizes all available compositions which are implemented as materials.

*Author contributions.* MK and MS wrote the manuscript. SZ and US edited the manuscript. SZ and US supervised the development of URANOS. MK designed the structure of URANOS, MK wrote and implemented the physics core of URANOS. MK and MS designed the graphical user interface and tested the software.

*Competing interests.* M. Köhli holds a CEO position at StyX Neutronica GmbH, Germany.

*Acknowledgements.* MK acknowledges all URANOS users who have contributed to the development of URANOS: Jannis Weimar, Paul Schattan, Daniel Rasche, Jannis Jakobi, Sugan Kanagasenthinathan, Patrick Stowell and Tatsuhiko Sato. MS and MK further acknowledge Peter Dietrich for funding and continuous support of the model development. URANOS was developed within several projects. Initial funding was provided by the projects 'Neutron Detectors for the MIEZE method' and 'Forschung und Entwicklung hochauflösender Neutronende-





**Table A1.** Available isotopes in URANOS and cross sections used, identifiers according to (Trkov et al., 2012).

| Isotope | Elastic | Inelastic | Absorption and others |
|---|---|---|---|
| $^1$H | MT=2 (MF=3, 4) | n/A | MT=5, 102, 208–210 |
| $^3$He | MT=2 (MF=3, 4) | n/A | MT=102, 103, 104 |
| $^{10}$B | MT=2 (MF=3, 4) | MT=51–54 | MT=107 |
| $^{11}$B | MT=2 (MF=3, 4) | MT=51–54 | MT=107 |
| $^{12}$C | MT=2 (MF=3, 4) | MT=51–58 | MT=5, 102, 103, 107 |
| $^{14}$N | MT=2 (MF=3, 4) | MT=51–60 | MT=5, 102–108, 208–210 |
| $^{16}$O | MT=2 (MF=3, 4) | MT=51–70 | MT=5, 102, 103, 105, 107, 208-210 |
| $^{19}$F | MT=2 (MF=3, 4) | MT=51–54 | MT=102, 103, 107 |
| $^{23}$Na | MT=2 (MF=3, 4) | MT=51–56 | MT=5, 102, 103, 107 |
| $^{27}$Al | MT=2 (MF=3, 4) | MT=51–58 | MT=5, 102, 103, 107, 208–210 |
| $^{28}$Si | MT=2 (MF=3, 4) | MT=51–58 | MT=5, 102, 103, 107, 208–210 |
| $^{32}$S | MT=2 (MF=3, 4) | MT=51–55 | MT=5, 102, 103, 107 |
| $^{35}$Cl | MT=2 (MF=3, 4) | MT=51–56 | MT=5, 102, 103, 107 |
| $^{39}$K | MT=2 (MF=3, 4) | MT=51–54 | MT=102, 103, 107 |
| $^{40}$Ar | MT=2 (MF=3, 4) | MT=51–55 | MT=5, 102, 103, 107, 208–210 |
| $^{48}$Ti | MT=2 (MF=3, 4) | MT=51–54 | MT=5, 102, 103, 107 |
| $^{52}$Cr | MT=2 (MF=3, 4) | MT=51–55 | MT=5, 102, 103, 107 |
| $^{53}$Cr | MT=2 (MF=3, 4) | MT=51–55 | MT=5, 102, 103, 107 |
| $^{55}$Mn | MT=2 (MF=3, 4) | MT=51–56 | MT=5, 102, 103, 107, 208–210 |
| $^{56}$Fe | MT=2 (MF=3, 4) | MT=51–58 | MT=5, 102, 103, 107, 208–210 |
| $^{58}$Ni | MT=2 (MF=3, 4) | MT=51–54 | MT=5, 102, 103, 107 |
| $^{63}$Cu | MT=2 (MF=3, 4) | - | MT=102 |
| $^{65}$Cu | MT=2 (MF=3, 4) | - | MT=102 |
| $^{155}$Gd | MT=2 (MF=3, 4) | MT=51–54 | MT=102 |
| $^{157}$Gd | MT=2 (MF=3, 4) | MT=51–54 | MT=102 |
| $^{206}$Pb | MT=2 (MF=3, 4) | MT=51–55 | MT=5, 102, 103, 107, 208–210 |
| $^{207}$Pb | MT=2 (MF=3, 4) | MT=51–55 | MT=5, 102, 103, 107, 208–210 |
| $^{208}$Pb | MT=2 (MF=3, 4) | MT=51–55 | MT=5, 102, 103, 107, 208–210 |

tektoren', funded by the German Federal Ministry for Research and Education (BMBF), grant identifier: 05K10VHA and 05K16PD1 and by
the DFG (German Research Foundation) research unit FOR 2694 Cosmic Sense via the project 357874777.



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



**Table B1.** List of preconfigured materials available in URANOS with their composition and density.

| Material | Density | Description |
|---|---|---|
| Helium | 0.125 kg/m$^3$ | $^3$He enriched gas |
| **Boron** | 2.34 g/cm$^3$ | 97 % $^{10}$B enriched |
| Boron natural | 2.46 g/cm$^3$ | 80.1 % $^{10}$B, 19.9 % $^{11}$B |
| Boron carbide | 2.42 g/cm$^3$ | $^{10}$B enriched $B_4C$ |
| Boron carbide | 2.51 g/cm$^3$ | $B_4C$ with natural boron |
| Boron trifluoride | 2.76 kg/m$^3$ | $^{10}$B enriched $BF_3$ gas |
| Methane | 0.656 kg/m$^3$ | $CH_4$ gas |
| Detector gas | 1.8 kg/m$^3$ | $ArCO_2$ gas (70:30, 80:20) |
| Aluminum | 2.66 g/cm$^3$ | |
| Aluminum oxide | 3.94 g/cm$^3$ | $Al_2O_3$ |
| Iron | 7.87 g/cm$^3$ | $^{56}$Fe |
| Steel (304L) | 8.03 g/cm$^3$ | with 68 % $^{56}$Fe, 16.3 % $^{52}$Cr, 2.7 % $^{53}$Cr, 9 % $^{58}$Ni, 2 % $^{28}$Si, 2 % $^{55}$Mn |
| Copper | 8.94 g/cm$^3$ | |
| Lead | 11.342 g/cm$^3$ | 24.1 % $^{206}$Pb, 22.1 % $^{207}$Pb, 52.4 % $^{208}$Pb |
| Salt | 2.16 g/cm$^3$ | NaCl |
| Diesel | 0.83 g/cm$^3$ | $CH_4$ |
| Graphite | 2.2 g/cm$^3$ | $^{12}$C |
| Gadolinium oxide | 7.41 g/cm$^3$ | $Gd_2O_3$ with 14.8 % $^{155}$Gd, 15.65 % $^{157}$Gd |
| **Polyethylene** | 0.95 g/cm$^3$ | HDPE, $CH_2$ |
| PE boronated | 0.95 g/cm$^3$ | HDPE with 3 % natural boron |
| **Polyimide** | 1.43 g/cm$^3$ | $C_{22}H_{10}N_2O_5$ |
| Quartz | 2.5 g/cm$^3$ | $SiO_2$ |
| **Stones** | 1.43 g/cm$^3$ | 75 % $SiO_2$, 25 % $Al_2O_3$ |
| **Water** | 1.0 g/cm$^3$ | $H_2O$ |
| **Soil** | >1.43 g/cm$^3$ | 50 % stones, (0-50) % water |
| **Air** | 1.2 kg/m$^3$ | 78 % $N_2$, 21 % $O_2$, 1 % Ar |
| Concrete | 2.0 g/cm$^3$ | 50 % stones, 10 % water |
| Cat litter | 1.1 g/cm$^3$ | 44 % H, 44 % O, 12 % Si |
| Trinitrotoluol | 1.654 g/cm$^3$ | 23.8 % H, 28.6 % O, 35 % C, 14.4 % N |
| Asphalt pavement | 2.58 g/cm$^3$ | 14 % H, 50 % O, 11 % C, 25 % Si |
| Plants | >2.2 kg/m$^3$ | 14 % H, 72 % O, 14 % C, plus air |
| Wood | 0.5 g/cm$^3$ | like plants |
| Snow new | 0.03 g/cm$^3$ | like water |
| Snow old | 0.3 g/cm$^3$ | like water |
| Ice | 0.85 g/cm$^3$ | like water |