# Peer review of "URANOS v1.0 - the Ultra Rapid Adaptable Neutron-Only Simulation for Environmental Research"

_Geoscientific Model Development, 2022_

## Author Comment (AC2)

**Response to the comments of the reviewers**

Dear Reviewer 3,

we are very grateful to the reviewer for the helpful comments and suggestions. In the following we address individually the comments to the manuscript 'URANOS v1.0 - the Ultra Rapid Adaptable Neutron-Only Simulation for Environmental Research' submitted to GMD. Reviewer's comments on the manuscript are bold, our answers italic and the latexdiff of the submitted paper indented in quotation.

Markus Köhli

Physikalisches Institut

Ruprecht-Karls-Universität Heidelberg

ANP-PAT

Martin Schön

Helmholtz Centre for Environmental Sciences, UFZ Leipzig

**▬ Reviewer 3**

**The publication shows a profound knowledge of the authors about the existing programs and the underlying physics and IT concepts. There are, however, some points that would improve the document:**

**For people from other fields of research it is partly difficult to read, because the methods are not always described, e.g. half a sentence on how CRNS works would be helpful.**

*We added to the introduction the following part:*

> The graphical user interface offers features specifically tailored to the needs of the field of Cosmic-Ray Neutron Sensing. The novel method retrieves subsurface soil moisture by measuring flux of cosmic-ray induced neutrons that scatter at the soil interface. With typical footprint ranges of hundreds of meters for stationary and beyond one kilometer for mobile sensors, it specifically addresses research questions in complex environments.

**Similarly, there are terms used that are not common to all fields. They should either be explained or alias names added, especially**
**- Is „evaporation" of neutrons the same as „spallation"of neutrons?**
**- Is „ray-casting" the same as „ray-tracing"?**

*The term spallation does not appear in the manuscript as currently it is not implemented due to its lack of relevance for environmental applications. We have clarified the term evaporation at its first occurrence:*

> such as evaporation, the delayed emission of MeV neutrons from excited nuclei

*We have added an explanation for ray casting:*

> Ray casting follows tracks from the source to the point of detection, contrary to ray tracing, which follows tracks backwards from the point of detection, but requires mostly deterministic interactions.

**Equation (1) to (4) are only consistent, if (3) and (4) calculate $p(x)\mathbf{d}p$, not $p(x)\mathbf{d}x$.**

*This statement is not clear to the authors. There might be a confusion here. $p(x)dx$ is $dp$, integrating $pdp$ would simply yield $[1/2\,p^2]$.*

**In Eq.(5), the second $\zeta$ should be replaced by anything else, e.g. $\zeta'$. As it is written now, the equation is not correct.**

*Yes, we changed the equation to an equivalent symbol.*

**Surprisingly, it sounds like the authors have doubts about the Monte Carlo method.**

*The authors might not have doubts about the method itself but about whether or not its application is justified. The Monte Carlo method is especially suitable for neutron calculations, but for other particle species like photons with a much higher flux it is probably not.*

**Some suggestion for text improvement:**
**- 47 - 49: can be omitted, as the programs are described in the following paragraphs?**

*We have removed the sentences.*

**- Chapter 1.2: it could be added that the programs dedicated to neutron instrumentation and virtual neutron experiments (McStas, VITESS, RESTRAX, ...) also allowed fast simulations by restricting its use to neutrons and ignoring nuclear reactions**

*We have added this proposition:*

> Restricting the calculation to neutrons and ignoring other nuclear reactions has been proven useful to increase computational speed in programs dedicated to neutron instrumentation and their representation in virtual experiments, like McStas (Lefmann and Nielsen, 1999), VITESS (Wechsler et al., 2000) and RESTRAX (Šaroun, J. and Kulda, J., 1997).

**- 106 - 112:: I cannot see the problem of the multigroup method**

*The multigroup method is motivated by the idea to describe the ensemble correctly, not individual neutrons. For example for providing criticality calculations. They are not only sensible to the thermal budget of the entity but also involve that the neutron flux changes the state of objects within the simulation time. As provided in the text, if neutrons undergo just one or two collisions it lacks the randomness, especially associated with outliers within the calculation.*

**- 119: performance of GEANT4: what is missing, speed or accuracy or ...?**

*We have added that information:*

> the computational speed of GEANT4 in typical scenarios is significantly lower than those of other codes.

**- 156 - 15: I don't understand that.**

*For thermal neutron energies, the Doppler broadening due to the relative velocity of neutron and its target influences the cross section as well. Not taking this into account leads to an incorrect scattering probability. We have added:*

> i.e. taking into account the Doppler effect.

**- 214f: Why? What is done instead?**

*Using ensemble statistics means to treat neutrons by analytical expressions like Fermi-age equations. Besides that in this highly complex task of CR neutrons the author does not even know whether such expressions can be derived out without mainly relying on perturbative calculations, ensemble statistics is useful for describing large numbers of particles. In case of typical neutron simulations this might not be the case. It also becomes complicated if certain effects are mainly due to kinematic outliers.*

**- 226: „Whereas"-> „While"**

*We have changed that.*

**Style: often there are too few commas in too long sentences.**

*We identified a number of long sentences and reduced their length.*

**Figure 1:**
**I wonder if neutrons are generated in the source or the soil layer.**
**An explanation of the particle symbols and a vertical scale would be good.**

*Both, their generation mostly takes place in the source layer, however, as indicated by the 'dot' at the evaporation label, neutrons can also be generated by other processes.*
*We have added the description of the particle symbols to the figure.*
*For the sake of the illustration the vertical scale has been compressed for the atmosphere and stretched for the ground. With both having a density difference of a factor of 1000, it is hard to provide an informative but still realistically scaled graph. We have added an annotation to the caption.*

**Figure 2:**
**- I think the „Layer stack" is still a „Neutron stack".**
**- Is there no flight direction stored?**

*The layer stack is the geometrical representation of the simulation domain, through which a neutron has to propagate. the neutron stack comprises neutrons with their initial (randomly generated) properties or neutrons which were generated by other processes during the simulation.*
*There is in fact a specific variable which only stores the flight direction (forward/backward) in order to facilitate some calculations. The figure does not explicitly mention the direction, however, it is included in the neutron vector.*

*Thank you very much for the review of our manuscript.*

---

## Author Comment (AC3)

**Response to the comments of the reviewers**

20.10.2022

Dear Reviewer 1,

we are very grateful to the reviewer for the helpful comments and suggestions. In the following we address individually the comments to the manuscript 'URANOS v1.0 - the Ultra Rapid Adaptable Neutron-Only Simulation for Environmental Research' submitted to GMD. Reviewer's comments on the manuscript are bold, our answers italic and the latexdiff of the submitted paper indented in quotation.

Markus Köhli

Physikalisches Institut

Ruprecht-Karls-Universität Heidelberg

ANP-PAT

Martin Schön

Helmholtz Centre for Environmental Sciences, UFZ Leipzig

**Reviewer 1**

**The authors present an excellent overview of the URANOS code in great detail. Given the continued use of CRNS this streamlined neutron transport code will be of great use to the community for evaluating and designing future experiments. The paper is well written and acceptable following some minor revisions. My only moderate comment is the need for more complex field data for validating the results (say irrigated row crops). These data would be greatly supported by URANOS model simulations to help untangle the complexity of this system. There have now been a number of papers developing this code (L600-620). Please see below for a few minor comments to address.**

*A full publication list has also been published in the GitHub repository:*
*https://github.com/mkoehli/uranos/blob/main/doc/PUBLICATIONS.md*
*Such validation studies are very time consuming both in terms of the simulation but also in terms of implementing representative validation measurements within the CRNS footprint and such evaluations require step-by-step simulation scenarios to properly understand the signal interpretation in the multidimensional parameter space. Adequate CRNS data for more complex irrigation situations are hardly available so far. In collaboration with FZ Jülich we are currently working on a paper on validating transport simulations in the context of irrigation on the sub-footprint scale. This manuscript summarizes the code structure and features and presents field data as examples of the working principle of the code, but does not intend them to be validations.*

**L58: Use of ontop is a bit awkward, please revise.**

*We simply deleted that word.*

**General comments:**

**L126: is $l$ the path length, please define here as well as sigma.**

*The information was unfortunately not provided in that context, we added it.*

> (a) the path length $l$, sampled from the probability of an interaction on a distance $dx$ in a homogeneous material of cross section $\Sigma$ using the random variable $r$, $l = -\ln(r)/\Sigma$, or (b) the thermal neutron velocity distribution.

**L252. „in a converter takes place". I don't follow this statement please revise.**

*We have rephrased that sentence.*

This process is called 'scoring'. It can be invoked when passing a specific volume or the track is terminated.

**L440. A space is needed between the words to start sentence.**

*That is a problem of font kerning of this template. It actually only optically appears to have a missing space character. There is one.*

**L498: Replace „As far as" with „If".**

*We have changed it.*

**L520: Figure 11 is discussed before figure 10.**

*We removed that sentence containing the reference in the introduction as the material for Fig. 11 is anyway discussed in the respective section.*

**L522: Neutron paths not neutrons paths.**

*We have changed it.*

**L585: For more than 50 years ...**

*We have correct it.*

**L604: fine tune.**

*We have correct it.*

**L600-620: I think a table summarizing the various studies with URANOS would be easier to read and reference for the reader. Please consider changing.**

*We agree with the reviewer that the current presentation does not provide the best overview possible. However, the published literature likewise is just a snapshot and does not qualify for representativeness. It represents the contexts in which URANOS has been used up to now, the list is simply currently expanding, even during submisson an revison of this manuscript.*

**L627. I don't follow this sentence about the evaluation of cross sections. Think there may be a missing word somewhere?**

*Thank you, there was one 'like' too much.*

Additionally, the performance depends on the chosen materials and energy range, as for elastic scattering or absorption only a few cross sections are evaluated.

*Thank you very much for the review of our manuscript.*

---

## Author Comment (AC4)

**Response to the comments of the reviewers**

20.10.2022

Dear Reviewer 2,

we are very grateful to the reviewer for the helpful comments and suggestions. In the following we address individually the comments to the manuscript 'URANOS v1.0 - the Ultra Rapid Adaptable Neutron-Only Simulation for Environmental Research' submitted to GMD. Reviewer's comments on the manuscript are bold, our answers italic and the latexdiff of the submitted paper indented in quotation.

Markus Köhli

Physikalisches Institut

Ruprecht-Karls-Universität Heidelberg

ANP-PAT

Martin Schön

Helmholtz Centre for Environmental Sciences, UFZ Leipzig

**▬ Reviewer 2**

**This is a well developed manuscript, demonstrating the applicability of a novel though well-tested code. The manuscript is well readable, inspirational and easy to access for environmental scientists although containting dense information on physical processes and modelling. I recommend publication in GMD with minor revisions urging the authors to expand on a few minor points (see below) and add a new section on limitations and outlook. This section will benefit the readers and inspire the next generation of researchers to build on this code, increase the user space and the capabilities of CRNS to deliver accurate environmental observations.**

**Please address following questions (in the new section):**
**What is the run time on a standard architecture (e.g. intel Core i5, 8GB RAM, Windows or Unix)?**
**Is the code parallelized for HPC applications?**
**What are the most expensive calculations and how to they scale from 2D to 3D for a „simple" geometric set up?**
**Can the atmosphere and cosmic ray interactions be modelled using URANOS code?**
**What is needed and what is the uncertainty to expand the code to include uncertainty from the cosmic ray energy spectrum at the top of the atmosphere and further particles?**
**Given URANOS is applied to simulate each CRNS locations, what are the remaining major uncertainties constraining the accuracy of cosmic ray neutron sensor derived hydrogen content in the CRNS footprint?**

*This comment on the proposition of a new section is not clear to the authors. Most of the questions brought up by the reviewer are addressed in section 9 'Performance benchmarks'. The first question on the run time is answered in detail providing several scenarios, including the standardized simple setup with just air and water. URANOS can be compiled an run under Linux/OSX but we did not carry out specific performance evaluations.*
*The scaling for an HPC architecture is not addressed in detail. There are some general issues for effective parallelization of Monte Carlo calculations, that means achieving a performance gain beyond running several instances in parallel. A Monte Carlo tool like URANOS calculates the history of neutrons subsequently. Each step features highly computationally unequal branches. Moreover, a neutron can terminate after one or 1000 calculations. This makes it problematic to parallelize on the neutron or the interaction level. Furthermore, it is necessary to constantly switch cross section files and geometry for which none of them directly fits into the cache of the CPU. The memory of a GPU would be suitably large but the complex interaction calculations do*

*not go along the capabilities of typical shader units. There would be alternative possibilities of beneficial realizations on a GPU, however, the coding task for such would be a job on its own. Historically other methods have been used to increase the performance of Monte Carlo simulations like the MultiGroup method or the calculation in fixed energy groups. URANOS itself scales well with the number of physical cores involved. Every additional hyper-threaded core contributes with the equivalent of half to one full core. We have added one sentence with respect to this question.*

*The performance evaluation is likewise answered in the respective section regarding the model complexity. In general the usage of voxels mainly blows up the memory usage, not the computational performance. In case the model domain contains a manifold of different materials the performance decreases slightly but not significantly, which is one of the benefits of a voxel engine. One of the most expensive calculation is evaluation of the scattering angle as for each energy a function of two interpolated Lagrange functions constructed by a set of coefficients has to be evaluated and then randomly sampled. However, this only comes into play for the MeV region. Thermal scattering is also computationally demanding as for each neutron a magnitude more of function evaluations is necessary than in the epithermal case. Summed up, there is not a single process which can be pointed out, which would be a significant bottle neck. This fact, however, is result of months of performance optimizations, a process in which each possible lengthy calculation has been identified.*

*As URANOS can only model neutrons, it features an effective high-energy cascade model. The physcial description of neutrons for the atmosphere (as a gas) is physically correct. The effective high-energy model is based on PHITS and MCNP calculations and emulates the presence of other particles and their typical neutron-generating effects. The mentioned other commonly used codes, however, do also not feature physical calculations of all particles and their respective processes. They break down typical possible interactions into analytical models and fine tune them to measured data. Moreover, many cross sections, especially of elements of high abundance in the environment, are missing, incorrect or feature high uncertainties. Therefore each code has its own methods to overcome those shortcomings. As to how much these models are appropriate representations is subject to ongoing studies. In order to provide at least a lower limit for uncertainties, we expand the example given in the manuscript: the best known cross section is the one associated with the elastic scattering of hydrogen, around 0.3 % statistical uncertainty. Other isotopes are in the range of 1 % or more. Inelastic scattering cross sections can easily reach 10 %. Considering such variations on the cross sections, typical quantities derived from a simulation has a relative error in the range of 1-3 %. Realistically speaking the absolute error of this kind of simulations can be estimated to lie in the order of 5-10 %. As, specifically speaking, CRNS measures relative differences the error on that lies significantly below that and is currently at least lower than the error on any other quantity used for comparisons between simulation and experiment, especially the soil moisture distribution horizontally and vertically within the footprint.*

**Line 132: It is unclear for the reader, how TRandom3 is programmed, language, and what a "modern architecture" is. Are the authors referring to HPC systems, GPU based HPCs or Laptops build in 2019? Please specify what you refer to as modern architecture ideally in the flops as measure for computational performance.**

*In order to provide you some 'extreme' examples: On a workstation CPU, Xeon E5-1620v2 from 2013 with 3.7 GHz scaling to 3.9 GHz, each call takes around 14 ns. On a Laptop CPU from 2020, Intel Core i5-1135G7 with 2.4 GHz scaling to 4.2 GHz (typically not reached), each call takes around 11 ns. This example shows that flops, even broken down to a single core operation, is not a good measure to characterize the performance here. The reasons for this behavior is that CPUs*

*are not simply 'number crunchers' and going into detail here on to what exactly such differences arise would go beyond the scope. The different call times shown in this example above will not influence the performance of URANOS significantly. Therefore we used the term 'modern architecture' to overcome that problem while still showing its performance. However, we additionally have provided a time range for the 'architecture' (CPU (< 5 years)).*

*In case the reader wants to know how TRandom3 is programmed and in which language, we have provided a reference to its implementation the framework ROOT, which is open source and therefore the code for TRandom3 can easily be accessed. In case the reader is interested in architectural considerations of the algorithm itself we provided a reference to the description of the Mersenne Twister.*

**Line 133: Please also mention possible other random number generators and are they available? If those are not relevant questions, then it seems the technical details are not needed and I recommend to simply state that „the TRandom3 random number generator is used" and remove the technical details on random number generation from the manuscript in this paragraph 2.2.**

*The random generator is the heart of a Monte Carlo code. Its working principles may not appear obvious to the user, but its technical details are relevant for experts. Although we agree with the reviewer that a detailed description of the use of random generators in Monte Carlo codes would go beyond the focus of the manuscript, we consider at least the specification of such basic features as speed and period length to be relevant. We want to show to the reader, that the performance is suitable for this application, with 10 ns per call other calculations tend to be the dominating factor. With the given period length and easily 1000 calls per neutron its randomness is assured for typical runs of 1e9 neutrons. By stating that it is seeded with the system time in ms, we indicate that by running several instances of URANOS started from the command line having the same initiation time, the randomness is not assured. ROOT features implementations of 9 different Random generators. Changing the random generator in URANOS is possible, one declaration at the program initiation needs to be changed, however, URANOS needs to be recompiled then.*

**Line 208: Please clarify what „MT numbers" are. Random „Mersenne Twister" numbers would not define reaction types, I assume.**

*In nuclear science both identifiers are common, in environmental science not. Sometimes in such interdisciplinary works, one faces the difficulty in standing at the interface between two different languages. Yet, we think that if the sentence „The ENDF format uses MT numbers to define reaction types and MF numbers to classify the data type of the respective set" is not descriptive enough the reference to the ENDF formats manual provided here is suitable to describe the MT and MF, material type and material file, numbers. In order to better guide the reader we have put both in apostrophes.*

The ENDF format uses the 'MT numbers' to identify neutron reaction types and 'MF numbers' to classify the data type of the respective file set.

**Line 215: What is ensemble statistics?**

*We have added:*

Such require a very large number of particles to derive laws without necessarily taking into account each individual state vector.

**Line 219: Please state which relevant and non-relevant interactions you are referring and what are the „two different types"?**

*We have added:*

(among them absorption, elastic and inelastic scattering as well as evaporation)

**Line 222: Please clarify, how is it possible that myons are not contributing while myons are the major cosmic rays entering the atmosphere, and neutrons are only a product of myon inter-action?**

*Thank you for that question. High-Energy (primary) neutrons are generated through several interaction channels. Evaporation neutrons are, however, mainly generated by other neutrons, protons and myons. The production probability decreases in the same order as the particle species is mentioned with myons contributing a few percent. Myons are the most abundant particles due to their low interaction probability (long mean free path). With a low interaction probability, the probability of producing neutrons is likewise low. In total one can describe it in the following way: the high energy cascade contributes to the production of neutrons by three different particle species with three different fractions and attenuation lengths (around $140\,cm^2/g$ for neutrons, around $110\,cm^2/g$ for protons and around $500\,cm^2/g$ for myons). Future work will address these different attenuation lengths and model the high-energy cascade accordingly.*

**Line 332: Please state recommended default starting angle.**

*There might be a confusion here. The source options listed in the respective chapter are 'artificial' sources not related to the cosmic neutron source. For artificial source definitions which are relevant for for example detector simulations you want to tailor the neutron beam accordingly. The cosmic-ray neutron source, which is discussed in the following section, is based on analytical functions. Here, the source angle cannot be specified as it is intrinsically set by the description by Sato.*

*Thank you very much for the review of our manuscript.*

---

## Author Response (AR2)

**Response to the comments of the reviewers**

26.11.2022

Dear Reviewer 2,

we are very grateful to the reviewer for the helpful comments and suggestions. In the following we address individually the comments to the manuscript 'URANOS v1.0 - the Ultra Rapid Adaptable Neutron-Only Simulation for Environmental Research' submitted to GMD. Reviewer's comments on the manuscript are bold, our answers italic and the latexdiff of the submitted paper indented in quotation.

Markus Köhli

Physikalisches Institut

Ruprecht-Karls-Universität Heidelberg

ANP-PAT

Martin Schön

Helmholtz Centre for Environmental Sciences, UFZ Leipzig

**▬ Reviewer 2**

**Thank you for addressing the comments.**
**A few minor points which I encourage to address in a minor revision:**

**The manuscript is confusing for the reader with up to 9 sections and not having the overview of the whole manuscript in one paragraph up front. Please add at the end of the introduction one paragraph on the structure of the manuscript, addressing briefly the name/content of each section.**

*We have added the suggestion.*

> The manuscript is divided into ten sections. After an overview about neutron Monte Carlo codes the physics concepts and the mathematical computation routines in URANOS are discussed in section 2. It is followed by a description of the design ideas of URANOS with the geometrical layer concept and its computational flow in section 4. The individual steps of the calculation are then presented in section 5 with the source configuration and in section 6 with the computation of neutron interactions. Finally section 7 discusses the scoring options. Section 2 to 7 therefore provide a description of the core of URANOS in terms of computation and design. In section 8 a variety of examples illustrates the capabilities of URANOS, the precision for typical use cases and provides links to previous research questions. Section 9 addresses the computational performance and section 10 presents the graphical user interface.

**L216: Please add, "...MT numbers (Material Type)" ... and "...MF numbers (Material File)...". The reader does not understand the MT or MF, if not explained. It could otherwise be removed straight away, if not important.**

*We considered and discussed the reviewer's suggestion. We think that the sentence "The ENDF format uses the 'MT numbers' to identify neutron reaction types and 'MF numbers' to classify the data type of the respective file set" is sufficient for that purpose. We tried to find official sources for describing the spelling out of those identifiers, not successfully though. The data structures for which those identifiers stand for are clearly described in the cited ENDF manual. In case we would have decided ourselves to simply call them A and B would not have changed the necessity for using them as representations for navigating though a data base. We also believe that the reviewer's demand to remove commonly used identifiers in case it would be impossible to spell them out, is not very well founded. It would practically mean to remove most of the variable declarations in the paper as we have not defined them literally. We hope that the reviewer might take into consideration that other fields of science have different standards.*

**L331: Not all reader read the manuscript from beginning to end. The name "cosmic ray neutron sensor" implies the space as source and not a random artificial layer above few meters above the soil surface. Add here the difference between neutron source (cosmic origin) and neutron source (URANOS source layer), although it was stated before.**

*Unfortunately it is not clear to us what the reviewer means with this comment. We tried to identify possible solutions. With the term "cosmic ray neutron sensor" the reviewer might have meant "cosmic ray neutron source". Line 331 is the beginning of the section about source options. It starts with a general introduction about source placement configuration and then discusses energy distributions. Later, the cosmic neutron source is discussed in its own section. Near the cited line 331 there is no such term as "cosmic ray neutron" appearing. Consecutively we have added the following statement to the section 'general sources':*

> The cosmic source used for studies of environmental neutrons is typically represented by a plane source.

And in the section of general sources we have added:

> Whereas the cosmic neutron source would typically be release from an extended layer, usually the geometry for other types of sources may be limited to a considerably small plane, scaling even down to a point source.

**Although addressed by the authors in the response letter, the minor importance of protons and myons to CNRS and this study has not been added to the manuscript. Please add where appropriate:**

**"The high energy cascade contributes to the production of neutrons by three different particle species with three different fractions and attenuation lengths (around 140 cm2/g for neutrons, around 110 cm2/g for protons and around 500 cm2/g for myons). Future work will address these different attenuation lengths and model the high-energy cascade accordingly."**

*We have added the suggestion to the section about the evaluation of the cosmic ray neutron spectra.*

> The high-energy cascade contributes to the production of neutrons by three different particle species with three different fractions and attenuation lengths (around $140\,\mathrm{cm}^2/\mathrm{g}$ for neutrons, around $110\,\mathrm{cm}^2/\mathrm{g}$ for protons and around $500\,\mathrm{cm}^2/\mathrm{g}$ for myons). Future work will address these different attenuation lengths and model the high-energy cascade accordingly.

*Thank you very much for the review of our manuscript.*

---

## Author Response (AR3)

**Response to the comments of the editor**

10.12.2022

Markus Köhli

Physikalisches Institut

Ruprecht-Karls-Universität Heidelberg

ANP-PAT

Martin Schön

Helmholtz Centre for Environmental Sciences, UFZ Leipzig

Dear Wolfgang Kurtz,

thank you very much for the helpful comments. In the following we address the comments to the manuscript 'URANOS v1.0 - the Ultra Rapid Adaptable Neutron-Only Simulation for Environmental Research' submitted to GMD. The editor's comments on the manuscript are in the regular font, our answers italic and the latexdiff of the submitted paper indented in quotation.

**■ Editor**

Dear authors,

thank you very much for submitting your revised manuscript to GMD. In my point of view, the comments of reviewer #2 were adequately addressed by your response and revisions. So from the content part, your paper is now ready for publication.

One remaining issue is the data availability. As already mentioned in the last iteration round, not only the model code but also the data/ examples used and presented in the manuscript must be made available via a public long-term archive (e.g. via Zenodo). Please consult the GMD code and data policy page (https://www.geoscientific-model-development.net/policies/code_and_data_policy.html) for detailed instructions about the requirements and make your data/ samples accessible accordingly.

If you have any further questions, do not hesitate to contact me directly.

Best regards,
Wolfgang Kurtz

*We might misunderstood the use of the code and data availability statement. Source code, examples, cross sections, model-dependent parameters and data for the exemplary plot are in the same repository. We have moved the code and data availability statements into one.*

> *Code and data availability.* The URANOS source code is made available at the Github repository https://github.com/mkoehli/uranos/tree/URANOS. URANOS v1.0 has been released under DOI: 10.5281/zenodo.6578668. This DOI represents all versions, and will always resolve to the latest one.
>
> Libraries and data used in this publication have been released in the above mentioned Github repository and are available from the Zenodo archive DOI: 10.5281/zenodo.6578668.
>
> The current model of URANOS is also available from the project website https://www.physi.uni-heidelberg.de/Forschung/ANP/Cascade/URANOS/.
>
> URANOS v1.0 is linked against ROOT 6.22.08, QT 5.15 and QCustomPlot 2.1.0.

*Thank you very much for the review of our manuscript.*

---

## Author Response (AR4)

**Response to the comments of the editor**

16.12.2022

Markus Köhli

Physikalisches Institut

Ruprecht-Karls-Universität Heidelberg

ANP-PAT

Martin Schön

Helmholtz Centre for Environmental Sciences, UFZ Leipzig

Dear Wolfgang Kurtz,

thank you very much for the helpful comments. In the following we address the comments to the manuscript 'URANOS v1.0 - the Ultra Rapid Adaptable Neutron-Only Simulation for Environmental Research' submitted to GMD. The editor's comments are in the regular font, our answers italic and the latexdiff of the submitted paper indented in quotation.

**▮ Editor**

Dear authors,

thank you very much for adding the data to the Zenodo repository.

Unfortunately, I found that the cited DOI points to the 'old' archive version that only contains the model code but not the data. Note also, that not the newest version of the code/data should be cited but the exact version that is described in the manuscript. From GMD's code and data policy page: „In every case, the citation from the paper must identify the exact version of the code and/or data used."

Please make sure that the cited DOI in the code data availability section references the exact version of the code AND data described in the manuscript. Ideally, you should also consider to synchronize the version number in the title with the one of the cited code.

Best regards,
Wolfgang Kurtz

*Actually the relevant data was already included in the original submission of the repository. The GitHub repository is to be understood also as a guide how to use the software. The data we added in the meantime are examples or results from other toolkits for comparison. The nature of this simulation is to assist neutron flux calculations, which means that it is able to generate most of the material shown in the manuscript with a minimum set of input parameters. The material we added improves the convenience for using the repository and carrying out own calculation, it is, however, not strictly necessary for yielding own results. Nevertheless, we created a new repository for source code, examples, cross sections, model-dependent parameters and data in the version v1.0 matching the manuscript title.*

> The URANOS source code is made available at the Github repository `https://github.com/mkoehli/uranos/tree/URANOS`. URANOS v1.0 has been released under DOI: 10.7910/DVN/THPNZW. Furthermore the code has been released including a collection of examples and use guides under DOI: 10.5281/zenodo.6578668. This DOI represents all versions, and will always resolve to the latest one.
>
> Libraries and data used in this publication have been released in the above mentioned Github repository and are available from the  Harvard dataverse archive DOI: 10.7910/DVN/THPNZW.

*Thank you very much for the review of our manuscript.*